# GlanceNets: Interpretable, Leak-proof Concept-based Models

**Emanuele Marconato**
Department of Computer Science
University of Pisa & University of Trento
Pisa, Italy
emanuele.marconato@unitn.it

**Andrea Passerini**
Department of Computer Science
University of Trento
Trento, Italy
andrea.passerini@unitn.it

**Stefano Teso**
Department of Computer Science
University of Trento
Trento, Italy
stefano.teso@unitn.it

## Abstract

There is growing interest in concept-based models (CBMs) that combine high-performance and interpretability by acquiring and reasoning with a vocabulary of high-level concepts. A key requirement is that the concepts be interpretable. Existing CBMs tackle this desideratum using a variety of heuristics based on unclear notions of interpretability, and fail to acquire concepts with the intended semantics. We address this by providing a clear definition of interpretability in terms of alignment between the model's representation and an underlying data generation process, and introduce GlanceNets, a new CBM that exploits techniques from disentangled representation learning and open-set recognition to achieve alignment, thus improving the interpretability of the learned concepts. We show that GlanceNets, paired with concept-level supervision, achieve better alignment than state-of-the-art approaches while preventing spurious concepts from unintentionally affecting its predictions. The code is available at https://github.com/ema-marconato/glancenet.

## 1 Introduction

Concept-based models (CBMs) are an increasingly popular family of classifiers that combine the transparency of white-box models with the flexibility and accuracy of regular neural nets [1–5]. At their core, all CBMs acquire a vocabulary of concepts capturing high-level, task-relevant properties of the data, and use it to compute predictions and produce faithful explanations of their decisions [6].

The central issue in CBMs is how to ensure that the concepts are *semantically meaningful* and *interpretable* for (sufficiently expert and motivated) human stakeholders. Current approaches struggle with this. One reason is that the notion of interpretability is notoriously challenging to pin down, and therefore existing CBMs rely on different heuristics—such as encouraging the concepts to be sparse [1], orthonormal to each other [5], or match the contents of concrete examples [3]—with unclear properties and incompatible goals. A second, equally important issue is *concept leakage*, whereby the learned concepts end up encoding spurious information about unrelated aspects of the data, making it hard to assign them clear semantics [7]. Notably, even concept-level supervision is insufficient to prevent leakage [8].

36th Conference on Neural Information Processing Systems (NeurIPS 2022).

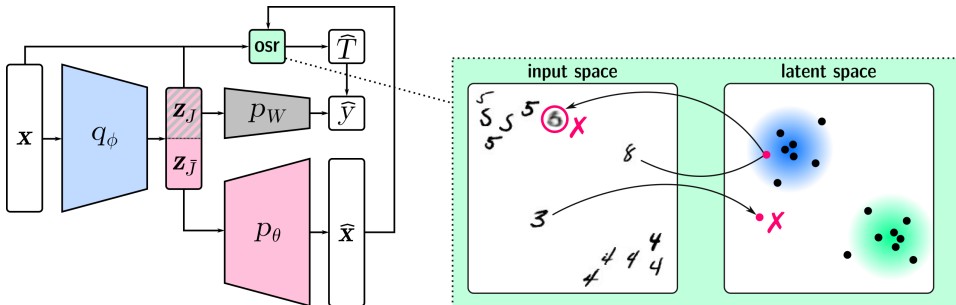

Figure 1: **Left**: Architecture of GlanceNets showing the encoder $q_\phi$, decoder $p_\theta$, classifier $p_W$, and open-set recognition step. **Right**: At test time, GlanceNets prevent leakage by identifying and rejecting out-of-distribution inputs using a combined strategy, shown here for a model trained on digits "4" and "5" only: the "3" is rejected as its embedding falls far away from prototypes of the two training classes (colored blobs), while the "8" is rejected as its reconstruction loss is too large.

Prompted by these observations, we define interpretability in terms of *alignment*: learned concepts are interpretable if they can be mapped to a (partially) interpretable data generation process using a transformation that preserves semantics. This is sufficient to unveil limitations in existing strategies, build an explicit link between interpretability and disentangled representations, and provide a clear and actionable perspective on concept leakage. Building on our analysis, we also introduce GlanceNets (aliGned LeAk-proof coNCEptual Networks), a novel class of CBMs that combine techniques from *disentangled representation learning* [9] and *open-set recognition* (OSR) [10] to actively pursue alignment – and guarantee it under suitable assumptions – and avoid concept leakage.

**Contributions:** Summarizing, we: (*i*) Provide a definition of interpretability as alignment that facilitates tapping into ideas from disentangled representation learning; (*ii*) Show that concept leakage can be viewed from the perspective of out-of-distribution generalization; (*iii*) Introduce GlanceNets, a novel class of CBMs that acquire interpretable representations and are robust to concept leakage; (*iv*) Present an extensive empirical evaluation showing that GlanceNets are as accurate as state-of-the-art CBMs while attaining better interpretability and avoiding leakage.

## 2   Concept-based Models: Interpretability and Concept Leakage

Concept-based models (CBMs) comprise two key elements: (i) A learned vocabulary of $k$ high-level concepts meant to enable communication with human stakeholders [11], and (ii) a simulatable [12] classifier whose predictions depend solely on those concepts. Formally, a CBM $f : \mathbb{R}^d \to [c]$, with $[c] := \{1, \ldots, c\}$, maps instances $\mathbf{x}$ to labels $y$ by measuring how much each concept activates on the input, obtaining an activation vector $\mathbf{z}(\mathbf{x}) := (z_1(\mathbf{x}), \ldots, z_k(\mathbf{x}))^T \in \mathbb{R}^k$, aggregating the activations into per-class scores $s_y(\mathbf{x})$ using a linear map [1, 3, 5], and then passing these through a softmax, i.e.,

$$s_y(\mathbf{x}) := \sum_j w_{yj} \cdot z_j(\mathbf{x}), \qquad p(y \mid \mathbf{x}) := \mathrm{softmax}(\mathbf{s}(\mathbf{x}))_y. \tag{1}$$

Each weight $w_{yj} \in \mathbb{R}$ encodes the relevance of concept $z_j$ for class $y$. The activations themselves are computed in a black-box manner, often leveraging pre-trained embedding layers, but learned so as to capture interpretable aspects of the data using a variety of heuristics, discussed below.

Now, *as long as the concepts are interpretable*, it is straightforward to extract human understandable local explanations disclosing how different concepts contributed to any given decision $(\mathbf{x}, y)$ by looking at the concept activations and their associated weights, thus abstracting away the underlying computations. This yields explanations of the form $\{(w_{yj}, z_j(\mathbf{x})) : j \in [k]\}$ that can be readily summarized[1] and visualized [13, 14]. Importantly, the score of class $y$ is conditionally independent from the input $\mathbf{x}$ given the corresponding explanation, i.e., $s_y(\mathbf{x}) \perp\!\!\!\perp \mathbf{x} \mid \mathcal{E}(\mathbf{x}, y)$, ensuring that the latter is faithful to the model scores. GlanceNets inherit all of these features.

**Heuristics for interpretability.** Crucially, CBMs are only interpretable insofar as their concepts are. Existing approaches implement special mechanisms to this effect, often pairing a traditional classification loss (such as the cross-entropy loss) with an auxiliary regularization term.

---

[1]For instance, by pruning those concepts that have little effect on the outcome to simplify the presentation.

Alvarez-Melis and Jaakkola [1] acquire concepts using an autoencoder augmented with a sparsification penalty encouraging distinct concepts to activate on different instances. Chen et al. [5] apply geometric transforms to learn mutually orthonormal concepts that thus encode complementary information and attain comparable activation ranges. These mechanisms – sparsity and orthogonality, respectively – alone cannot prevent capturing features that are not semantic in nature.

A second group of CBMs tackle this issue by constraining the concepts to match *concrete* cases, in the hope that these are better aligned with human intuition [15]. For instance, prototype classification networks [2], part-prototype networks [3], and related approaches [16–18] model concepts using prototypes in embedding space that perfectly match training examples or parts thereof. Depending on the embedding space, which ultimately determines the distance to the prototypes, concepts learned this way may activate on elements unrelated to the example they match, leading to unclear semantics [19].

Closest to our work, concept bottleneck models (CBNMs) [20, 4] align the concepts using concept-level supervision – possibly obtained from a separate source, like ImageNet [21] – either sequentially or in tandem with the top-level dense layer. From a statistical perspective, this seems perfectly sensible: if the supervision is unbiased and comes in sufficient quantity, and the model has enough capacity, this strategy *appears* to guarantee the learned and ground-truth concepts to match.

**Concept leakage in concept-bottleneck models.** Unfortunately, concept-level supervision is *not* sufficient to guarantee interpretability. Mahinpei et al. [7] have demonstrated that concepts acquired by CBNMs pick up spurious properties of the data. In their experiment, they learn two concepts $z_4$ and $z_5$, meant to represent the 4 and 5 MNIST digits, using concept-level supervision, and then show that – surprisingly – these concepts can be used to classify all *other* digits (i.e., MNIST images that are neither 4's nor 5's) as even or odd significantly better than random guessing. This phenomenon, whereby learned concepts unintentionally capture information about unobserved concepts, is known as *concept leakage*.

Intuitively, leakage occurs because in CBNMs the concepts end up unintentionally capturing distributional information about unobserved aspects of the input, failing to provide well-defined semantics. However, a clear definition of leakage is missing, and so are strategies to prevent it. In fact, separating concept learning from classification and increasing the amount of supervision for the observed concepts (here, 4 and 5) is not enough [8]. A key contribution of our paper is showing that leakage can be understood from the perspective of domain shift and dealt with using open-set recognition [10].

## 3 Disentangling Interpretability and Concept Leakage

The main issue with heuristics used by CBMs is that they are based on unclear notions of interpretability. In order to develop effective algorithms, we propose to view interpretability as a form of *alignment* between the machine's representation and that of its user. This enables us to identify conditions under which interpretability can be achieved, build links to well-understood properties of representations, and leverage state-of-the-art learning strategies.

**Interpretability.** We henceforth focus on the (rather general) generative process shown in Fig. 2: the observations $\mathbf{X} \in \mathbb{R}^d$ are caused by $n$ generative factors $\mathbf{G} \in \mathbb{R}^n$, themselves caused by a set of confounds $\mathbf{C}$ (including the label $Y$ [22]). Notice that the generative factors *can* be statistically dependent due to the confounds $\mathbf{C}$, but as noted by Suter et al. [23], the total causal effect [24, Def. 6.12] between $G_i$ and $G_j$ is zero for all $i \neq j$. The generative factors capture all information necessary to determine the observation [23, 25], so the goal is to learn concepts $\mathbf{Z} \in \mathbb{R}^k$ that recover them. The variable $T$ is also a confounding factor, but it is kept separate from $\mathbf{C}$ as it relates to concept leakage, and will be formally introduced later on.

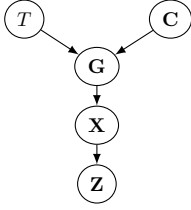

Figure 2: The data generation process.

We posit that a (learned) representation is only interpretable if it supports *symbolic communication* between the model and the user, in the sense that it shares the same (or similar enough) semantics to the user's representation. The latter is however generally unobserved. We therefore make a second, critical assumption that *some* of the generative factors $\mathbf{G}_I \subseteq \mathbf{G}$ are interpretable to the user, meaning that they can be used as a proxy for the user's internal representation. Naturally, not all generative factors are interpretable [26], but in many applications some of them are. For instance,

in dSprites [27] the generative factors encode the position, shape and color of a 2D object, and in CelebA [28] the hair color and nose size of a celebrity. Human observers have a good grasp of such concepts.

**Interpretability as alignment.** Under this assumption, if the variables $\mathbf{Z}_J \subseteq \mathbf{Z}$ are *aligned* to the generative factors $\mathbf{G}_I$ by a map $\alpha : \mathbf{g} \mapsto \mathbf{z}_J$ that preserves semantics, they are themselves interpretable. Now, defining what a semantics-preserving map should look like is challenging, but constructing one is not: the identity is clearly one such map, and so are maps that permute the indices and independently rescale the individual variables. One desirable property is that $\alpha$ does not "mix" multiple $G$'s into a single $Z$. E.g., if $Z$ blends together head tilt, hair color, and nose size, users will have trouble pinning down what it means.[2] This property can be formalized in terms of *disentanglement* [29, 23, 9]. This is however insufficient: we wish the map between $G_i$ and its associated factor $Z_j$ to be "simple", so as to *conservatively* guarantee that it preserves semantics. This makes alignment strictly stronger than disentanglement.

Motivated by these desiderata, we say that $\mathbf{Z}_J$ is *aligned* to $\mathbf{G}_I$ if it satisfies:

(i) **Disentanglement.** There exists an injective map between indices $\pi : [n_I] \to [k]$, where $[n_I]$ identifies the subset of generative factors indexes in $\mathbf{G}_I$, such that, for all $i, i' \in [n_I]$, $i \neq i'$, and $j = \pi(i)$, it holds that fixing $G_i$ is enough to fix $Z_j$ regardless of the value taken by the other generative factors $G_{i'}$, and

(ii) **Monotonicity.** The map $\alpha$ can be written as $\alpha(\mathbf{g}) = (\mu_1(g_{\pi(1)}), \ldots, \mu_n(g_{\pi(n_I)}))^T$, where the $\mu_i$'s are monotonic transformations. This holds, for instance, for linear transformations of the form $A \left(g_{\pi(1)}, \ldots, g_{\pi(n_I)}\right)^T$, where $A \in \mathbb{R}^{n_I \times k}$ is a matrix with no non-zero off-diagonal entries. This second requirement can be relaxed depending on the application.

Notice that we do not require each $G_i$ to map to a *single* $Z_j$ (a property known as *completeness* [29]): $\mathbf{Z}_J$ is interpretable even if it contains multiple – perhaps slightly different, but aligned – transformations of the same $G_i$.

**Measuring alignment with DCI.** Disentanglement can be measured in a number of ways [30], but most of them provide little information about how simple the map $\alpha$ is. In order to estimate alignment, we repurpose DCI, a measure of disentanglement introduced by Eastwood and Williams [29], see also **??**. According to this metric, a representation $\mathbf{Z}_J$ is disentangled if there exists a regressor that, given $\mathbf{z}_J$, can predict $\mathbf{g}_I$ with high accuracy using few $z_i$'s to predict each $g_i$. Following [29], we use a linear regressor with parameters $B \in \mathbb{R}^{k \times n_I}$ on the test set – assuming that it is annotated with the interpretable generative factors and corresponding learned representations – and then measure how diffuse the weights associated to each latent factor are. We do this by normalizing them and computing their average Shannon entropy over all $G_i$'s, i.e.,

$$- \sum_{j \in [k]} \rho_j \left( \sum_{i \in [n_I]} \bar{b}_{ji} \log \bar{b}_{ji} \right), \quad \text{where } \bar{b}_{ji} = b_{ji} / \sum_{j' \in [k]} b_{j'i} \quad \text{and} \quad \rho_j = \sum_i b_{ji} / \sum_{j'i} b_{j'i} \tag{2}$$

Hence, DCI gauges the degree of mixing that a linear map can attain using the learned representation $\mathbf{Z}$, and as such it indirectly measures alignment, with $B$ approximating the inverse of $A$.

**Achieving alignment with concept-level supervision.** It has been shown that disentanglement cannot be achieved in the purely unsupervised setting [31]. This immediately entails that alignment is also impossible in that setting, highlighting a core limitation of approaches like self-explainable neural networks [1]. However, disentanglement can be attained if supervision about the generative factors is available, even only for a small percentage of the examples [32]. As a matter of fact, supervision is used in representation learning to achieve *identifiability*, a stronger condition than – and that entails both of – disentanglement *and* alignment [33]. Thus, following CBNMs, we seek alignment by leveraging concept-level supervision.

**Interpretability and concept leakage.** Intuitively, concept leakage occurs when a model is trained on a data set on which:

(i) Some generative factors $\mathbf{G}_V \subset \mathbf{G}$ **vary**, while the others $\mathbf{G}_F = \mathbf{G} \setminus \mathbf{G}_V$ are **fixed**, and

---

[2]The converse is not true: interpretable concepts with *compatible* semantics can be mixed without compromising interpretability. E.g., rotating a coordinate system gives another intuitive coordinate system. Our point is that conservatively avoiding mixing helps to preserve semantics.

(*ii*) The two groups of factors are **statistically dependent**.

For instance, in the even vs. odd experiment $4$ and $5$ play the role of $\mathbf{G}_V$ and the other digits of $\mathbf{G}_F$. CBNMs with access to supervision on $\mathbf{G}_V$ tend to acquire a latent representation that approximates these factors. But, because of (*ii*), this representation correlates with the fixed factors $\mathbf{G}_F$. This immediately explains why additional supervision on $\mathbf{G}_V$ cannot prevent leakage, but rather has the opposite effect: the better a latent representation matches $\mathbf{G}_V$, the more information it conveys about $\mathbf{G}_F$.

In contrast with previous assessments [7, 8], we observe that this phenomenon can be viewed as a special form of domain shift: the training examples are sampled from a ground-truth distribution $p(\mathbf{X}, \mathbf{G} \mid T = 1)$ in which $\mathbf{G}_F$ is approximately fixed, e.g., $p(\mathbf{G}_F \mid T = 1) = \delta(\mathbf{g}'_F)$ for some vector $\mathbf{g}'_F$, while in the test set, the data is sampled from a different distribution $p(\mathbf{X}, \mathbf{G} \mid T = 0)$ in which $\mathbf{G}_F$ is no longer fixed. In the MNIST task, for instance, when $T = 1$ no concept besides $4$ and $5$ can occur, while all concepts *except* $4$ and $5$ can occur when $T = 0$. Here, $T \in \{0, 1\}$ selects between training and test distribution, see Fig. 2. Now, CBMs have no strategy to cope with domain shift and thus cannot disambiguate between known training and unknown test concepts.

Motivated by this, we propose then to tackle concept leakage by designing a CBM specifically equipped with strategies for detecting – at *inference* time – instances that do not belong to the training distribution using OSR [10]. The idea is to estimate the value of the variable $T$ at inference time, essentially predicting whether an input was sampled from a distribution similar enough to the training distribution, and therefore can be handled by a model learned on this distribution, or not. This strategy proves very effective in practice, as shown by our empirical evaluation (Section 5.2).

## 4   Addressing Alignment and Leakage with GlanceNets

GlanceNets combine a VAE-like architecture [34, 35] for learning disentangled concepts with a prior and classifier designed for open-set prediction [36]. In order to accommodate for non-interpretable factors, the latent representation of GlanceNets $\mathbf{Z}$ is split into two: (i) $k$ concepts $\mathbf{Z}_J$, aligned to the *interpretable* generative factors $\mathbf{G}_I$, that are used for prediction, and (ii) $\bar{k}$ *opaque* factors $\mathbf{Z}_{\bar{J}}$ that are only used for reconstruction. Specifically, a GlanceNet comprises an encoder $q_\phi(\mathbf{Z} \mid \mathbf{X})$ and a decoder $p_\theta(\mathbf{X} \mid \mathbf{Z})$, both parameterized by deep neural networks, as well as a classifier $p_W(Y \mid \mathbf{Z}_J)$ feeding off the interpretable concepts only. The overall architecture is shown in Fig. 1.

Following other CBMs, the classifier is implemented using a dense layer with parameters $W \in \mathbb{R}^{v \times k}$ followed by a softmax activation, i.e., $p_W(Y \mid \mathbf{z}_J) := \mathrm{softmax}(W\mathbf{z}_J)$, and the most likely label is used for prediction. The class distribution is obtained by marginalizing over the encoder's distribution:

$$p(Y \mid \mathbf{x}) := \mathbb{E}_{q_\phi(\mathbf{z}|\mathbf{x})}[p(Y \mid \mathbf{z}, \mathbf{x})] = \mathbb{E}_{q_\phi(\mathbf{z}|\mathbf{x})}[p_W(Y \mid \mathbf{z}_J)] \tag{3}$$

Equality holds because $Y \perp\!\!\!\perp \mathbf{X} \mid \mathbf{Z}_J$. In order to expedite the computation, we follow the general practice of approximating the integral as $\mathrm{softmax}(W \, \mathbb{E}_{q_\phi(\mathbf{z}|\mathbf{x})}[\mathbf{z}_J]) = \mathrm{softmax}(W[\boldsymbol{\mu}_\phi(\mathbf{x})]_J)$.

In contrast to regular VAEs, GlanceNets associate each class to a prototype in latent space through the prior $p(\mathbf{Z} \mid \mathbf{Y})$, which is conditioned on the class and modelled as a *mixture of gaussians* with one component per class. The encoder, decoder, and prior are fit on data so as to maximize the evidence lower bound (ELBO) [37], defined as $\mathbb{E}_{p_D(\mathbf{x},y)}[\mathcal{L}(\theta, \mathbf{x}, y; \beta)]$ with:

$$\mathcal{L}(\theta, \mathbf{x}, y; \beta) := \mathbb{E}_{q_\phi(\mathbf{z}|\mathbf{x})}[\log p_\theta(\mathbf{x} \mid \mathbf{z}) + \log p_W(y \mid \mathbf{z}_J)] - \beta \cdot \mathsf{KL}(q_\phi(\mathbf{z} \mid \mathbf{x}) \,\|\, p(\mathbf{z} \,|\, y)) \tag{4}$$

Here, $p_D(\mathbf{x}, y)$ is the empirical distribution of the training set $D = \{(\mathbf{x}_i, y_i) : i = 1, \ldots, m\}$. The first term of Eq. (4) is the likelihood of an example after passing it through the encoder distribution.

The second term penalizes the latent vectors based on how much their distribution differs from the prior and encourages disentanglement. As mentioned in Section 3, learning disentangled representations is impossible in the unsupervised i.i.d. setting [31]. Following Locatello et al. [32], and similarly to CBNMs, we assume access to a (possibly separate) data set $\tilde{D} = \{(\mathbf{x}_\ell, \mathbf{g}_{I,\ell})\}$ containing supervision about the *interpretable* generative factors $\mathbf{G}_I$ and integrate it into the ELBO by replacing the per-example loss $\mathcal{L}$ in Eq. (4) with:

$$\mathbb{E}_{p_\mathcal{D}(\mathbf{x},y)}\big[\mathcal{L}(\theta, \mathbf{x}, y; \beta)\big] + \gamma \cdot \mathbb{E}_{p_{\tilde{D}}(\mathbf{x},\mathbf{g})}\mathbb{E}_{q_\phi(\mathbf{z}|\mathbf{x})}\big[\Omega(\mathbf{z}, \mathbf{g})\big] \tag{5}$$

where $\gamma > 0$ controls the strength of the concept-level supervision. Following [32], the term $\Omega(\mathbf{z}, \mathbf{g})$ penalizes encodings sampled from $q_\phi(\mathbf{z} \,|\, \mathbf{x})$ for differing from the annotation $\mathbf{g}$. Specifically, we implement this term using the average cross-entropy loss $\Omega(\mathbf{z}, \mathbf{g}) := -\sum_k g_k \log \sigma(z_k) + (1 - g_k) \log(1 - \sigma(z_k))$, where the annotations $g_k$ are rescaled to lie in $[0, 1]$ and $\sigma$ is the sigmoid function.

**Dealing with concept leakage.** In order to tackle concept leakage, GlanceNets integrate the OSR strategy of Sun et al. [36], indicated in Fig. 1 by the "osr" block. This strategy identifies out-of-class inputs by considering the class prototype $\mu_y := \mathbb{E}_{p(\mathbf{z}|y)}[\mathbf{z}]$ in $\mathbb{R}^k$ defined by the prior distribution and the decoder $p_\theta(\mathbf{x}|\mathbf{z})$. Recall that the prior is fit jointly with the encoder, decoder, and classifier by optimizing the ELBO. Once learned, a GlanceNet uses the training data to estimate: (*i*) a distance threshold $\eta_y$, which defines a spherical subset in the latent space $\mathcal{B}_y = \{\mathbf{z} : \|\mu_y - \mathbf{z}\| < \eta_y\}$ centered around the prototype of class $y$ (i.e., the mean of the corresponding Gaussian mixture component), and (*ii*) a maximum threshold on the reconstruction error $\eta_{thr}$. If new data points have reconstruction error above $\eta_{thr}$ or they do not belong to any subset $\mathcal{B}_y$, they are inferred as open-set instances, i.e., $\hat{T} = 0$. In practice, we found that choosing the thresholds as to include 95% of training examples to work well in our experiments. Further details are available in Appendices **??** and **??**.

### 4.1 Benefits and Limitations

GlanceNets can naturally be combined with different VAE-based architectures for learning disentangled representations [38], including $\beta$-TCVAEs [39], InfoVAEs [40], DIP-VAEs [41], and JL1-VAEs [42]. Since our experiments already show substantial benefits for GlanceNets building on $\beta$-VAEs [43], we leave a detailed study of these extensions to future work.

Like CBNMs, GlanceNets foster alignment by leveraging supervision on the interpretable generative factors [32], possibly derived from an external data set [20]. However, GlanceNets can be readily adapted to a variety of different kinds of supervision used for VAE-based models, including *partially* annotated examples [26], group information [44], pairings [45, 46] and other kinds of weak supervision [47, 48], as well as feedback from a domain expert [49]. On the other hand, CBNMs are incompatible with these approaches.

One limitation inherited from VAEs by GlanceNets is the assumption that the interpretable generative factors are disentangled from each other [23]. In practice, GlanceNets work even when this does not hold (as in our even vs. odd experiment, see Section 5.2). However, one direction of future work is to integrate ideas from hierarchical disentanglement [50].

## 5 Empirical Evaluation

In this section, we present results on several tasks showing that GlanceNets outperform CBNMs [20] in terms of alignment and robustness to leakage, while achieving comparable prediction accuracy. All experiments were implemented using Python 3 and Pytorch [51] and run on a server with 128 CPUs, 1TiB RAM, and 8 A100 GPUs. GlanceNets were implemented on top of the `disentanglement-pytorch` [52] library. All alignment and disentanglement metrics were computed with `disentanglement_lib` [31]. Code for the complete experimental setup is available on GitHub at the link: https://github.com/ema-marconato/glancenet. Additional details on architectures and hyperparameters can also be found in the Supplementary Material.

### 5.1 GlanceNets achieve better alignment than CBNMs

In a first experiment, we compared GlanceNets with CBNMs on three classification tasks for which supervision on the generative factors is available. In order to evaluate the impact of this supervision on the different competitors, we varied the amount of training examples annotated with it from $1\%$ to $100\%$. For each increment, we measured prediction performance using accuracy, and alignment using the linear variant of DCI [29] discussed in Section 3.

**Data sets.** We carried out our evaluation on two data sets taken from the disentanglement literature and a very challenging real-world data set. *dSprites* [27] consists of $64 \times 64$ black-and-white images of sprites on a flat background, where each sprite is determined by one categorical and four generative factors, namely `shape`, `size`, `rotation`, `position_x`, and `position_y`. The images were obtained by discretizing and enumerating the generative factors, for a total of $3 \times 6 \times 40 \times 32 \times 32$

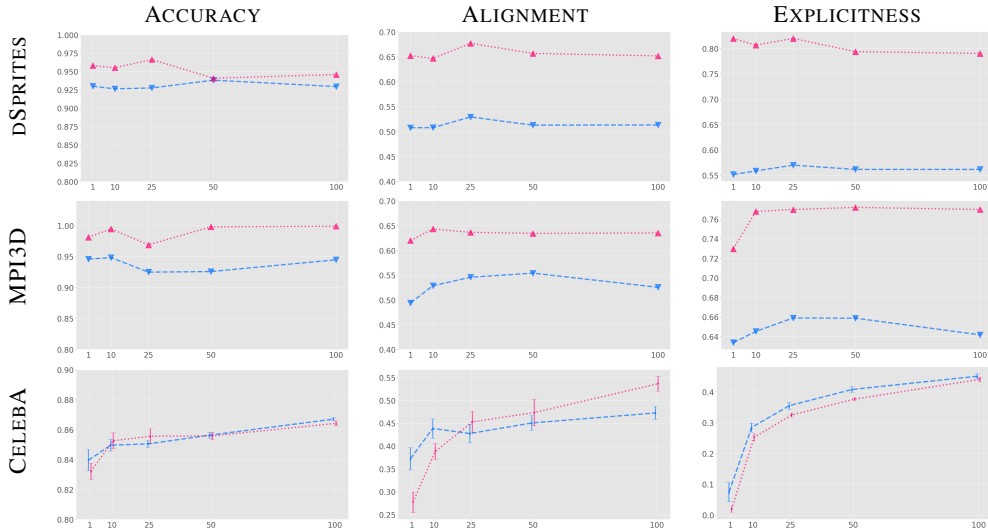

Figure 3: **GlanceNets are better aligned than CBNMs.** Each row is a data set and each column reports a different metric. The horizontal axes indicate the % of training examples for which supervision on the generative factors is provided. Remarkably, in all data sets **GlanceNets** achieve substantially better alignment than **CBNMs** for the same amount of supervision, and achieve comparable accuracy in 14 cases out of 15.

images. *MPI3D* [53] consists of $64 \times 64$ RGB rendered images of 3D shapes held by a robotic arm. The generative factors are `object_color`, `object_shape`, `object_size`, `camera_height`, `background_color`, and the `horizontal` and `vertical` position of the arm. The data contains $6 \times 6 \times 2 \times 3 \times 4 \times 40 \times 40$ examples. *CelebA* [28] is a collection of $178 \times 218$ RGB images of over 10k celebrities, converted to $64 \times 64$ by first cropping them to $178 \times 178$ and then rescaling. Images are annotated with 40 binary generative factors including hair color, presence of sunglasses, *etc*. Since we are interested in measuring alignment, we considered only those 10 factors that CBNMs can fit well (in the Appendix). We also dropped all those examples for which hair color is not unique (e.g., annotated as both `blonde` and `black`), obtaining approx. $127k$ examples. CelebA is more challenging than dSprites and MPI3D, as it does not include all possible factor variations and the generative factors – although disentangled – are insufficient to completely determine the contents of the images. For dSprites and MPI3D, we used a random 80/10/10 train/validation/test split, while for CelebA we kept the original split [28].

We generated the ground-truth labels $y$ as follows. For dSprites, we labeled images according to a random but fixed linear separator defined over the *continuous* generative factors, chosen so as to ensure that the classes are balanced. For MPI3D and CelebA, we focused on the *categorical* factors instead. Specifically, we clustered all images using the algorithm of [54], for a total of 10 and 4 clusters for MPI3D and CelebA respectively, and then labeled all examples based on their reference cluster. This led to slightly unbalanced classes containing different percentages of examples, ranging from 5% to 16% in MPI3D and from 21% to 29% in CelebA.

**Architectures.** For dSprites and MPI3D, we implemented the encoder as a six layer convolutional neural net, while for CelebA we adapted the convolutional architecture of Ghosh et al. [55]. We employed a six layer convolutional architecture for the decoder in all cases, for simplicity, as changing it did not lead to substantial differences in performance. In all cases, as for all CBMs (see Section 2), the classifier was implemented as a dense layer followed by a softmax activation. The very same architectures were used for both GlanceNets and CBNMs, for fairness. For each data set, we chose the latent space dimension as the total number of generative factors, where categorical ones are one hot encoded. In particular, we used 7 latent factors for dSprites, 21 for MPI3D and 10 for CelebA. Further details are included in the Supplementary Material.

**Results and discussion.** The results of this first experiment are reported in Fig. 3. The behavior of both competitors on dSprites and MPI3D was extremely stable, owing to the fact that these data sets cover an essentially exhaustive set of variations for all generative factors, so we report their hold-out

performance on the test set. Since for CelebA variance was non-negligible, we ran both methods 7 times varying the random seed used to initialize the network and report the average performance across runs and its standard deviation.

In addition to alignment, we also report explicitness [29], which measures how well the linear regressor employed by DCI fits the generative factors. The higher, the better. Details on its evaluation are included in Suppl. Material.

The plots clearly show that, although the two methods achieve high and comparable accuracy in all settings, GlanceNets attain better alignment in all data sets and for all supervision regimes than CBNMs, with a single exception in CelebA using low values of supervision, for a total of 13 wins out of 15 cases. In all *disentanglement* data sets, there is a clear margin between the alignment achieved by GlanceNets and that of CBNMs: performances vary up to maximum of 15% in dSprites, and a minumum of 8% in MPI3D. In CelebA, the gap is evident with full supervision (almost 8% of difference in alignment), and GlanceNets still attain overall better scores in the 25% and 50% regime. On the other hand, performance are lower, but comparable, with 10% supervision. The case at 1% refers to an extreme situation where both CBNMs and GlanceNets struggle to align with generative factors, as is clear also from the very low explicitness.

In dSprites and MPI3D, both GlanceNets and CBNMs quickly achieve very high alignment at 1% supervision, as expected [32], whereas better results in CelebA are obtained with growing supervision. Also, both models display similar stability on this data set, as shown by the error bars in the plot.

### 5.2 GlanceNets are leak-proof

Next, we evaluated robustness to concept leakage in two scenarios that differ in whether the un-observed generative factors are disentangled with the observed ones or not, see Section 3. In both experiments, we compare GlanceNets with a CBNM and a modified GlanceNet where the OSR component has been removed (denoted CG-VAE).

**Leakage due to unobserved entangled factors.** We start by replicating the experiment of Mahinpei et al. [7]: the goal is to discriminate between even and odd MNIST images using a latent representation $\mathbf{Z} = (Z_4, Z_5)$ obtained by training (with complete supervision on the generative factors) *only* on examples of 4's and 5's. Leakage occurs if the learned representation can be used to solve the prediction task better than random on a test set where all digits except 4 and 5 occur.[3] During training, we use the digit label for conditioning the prior $p(\mathbf{Z} \mid \mathbf{Y})$ of the GlanceNet. More qualitative results are collected in **??**.

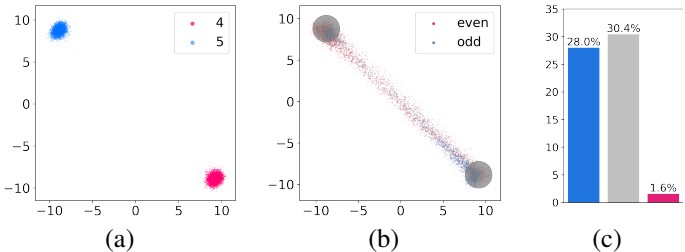

Figure 4: **GlanceNets are leak-proof on MNIST.** (*a*) Training set embedded by GlanceNet with $\beta = 100$; axes indicate $z_4$ and $z_5$ and color the concept label, i.e., 4 vs. 5. (*b*) Latent representations of the test images, divided in even vs. odd. Every ball in light gray denotes the region $|\mu_y - \mathbf{z}| < \eta_y$ for each class prototype $y$. For more details, refer to Section 3. (*c*) Information Leakage performances of the considered models: CBNM, CG-VAE and GlanceNet.

Fig. 4 (a, b) illustrates the latent representations of the training and test set output by a GlanceNet: since the two digits are mutually exclusive, the model has learned to map all instances along the $(z_4, z_5)$ diagonal. This is where OSR kicks in: if an input is identified as open-set, $T$ is predicted as 0 by the OSR component and the input is rejected. In all leakage experiments, we implement rejection by predicting a random label. Since MNIST is balanced, we measure leakage by computing

---

[3]Margeloiu et al. [8] perform classification using a multi-layer perceptron on top of $\mathbf{z}$. Following the CBM literature, we use a linear classifier instead. Leakage occurs regardless.

the difference in accuracy between the classifier and an ideal random predictor, i.e., $2 \cdot |\text{acc} - \frac{1}{2}|$: the smaller, the better. The results, shown in Fig. 4 (c), show a substantial difference between GlanceNet and the other approaches. Consistently with the values reported in [7], CBNMs are affected by a considerable amount of leakage, around $28\%$. This is not the case for our GlanceNet: most (approx. $85\%$) test images are correctly identified as open-set and rejected, leading to a very low (about $2\%$) leakage, $26\%$ less than CBNMs. The results for CG-VAE also indicate that removing the open-set component from GlanceNets dramatically increases leakage back to around $30\%$. This shows that alignment and disentanglement alone are not sufficient, and that the open-set component plays a critical role for preventing leakage.

**Leakage due to unobserved disentangled factors.** Next, we analyze concept leakage between *disentangled* generative factors using the dSprites data set. To this end, we defined a binary classification task in which the ground-truth label depends on `position_x` and `position_y` only. In particular, instances within a fixed distance from $(0, 0)$ are annotated as positive and the rest as negative, as shown in Fig. 5 (a). In order to trigger leakage, all competitors are trained (using full concept-level supervision, as before) on training images where `shape`, `size` and `rotation` vary, but `position_x` and `position_y` are almost constant (they range in a small interval around $(0.5, 0.5)$, cf. Fig. 5). leakage occurs if the learned model can successfully classify test instances where `position_x` and `position_y` are no longer fixed. More qualitative results can be found in **??**.

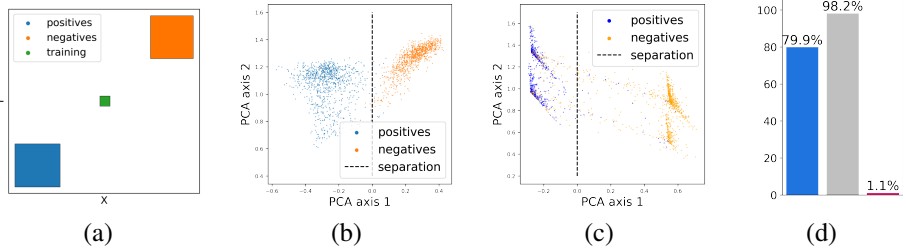

| (a) | (b) | (c) | (d) |

Figure 5: **GlanceNets are leak-proof on dSprites.** (*a*) The variations over `pos_x` and `pos_y` for the training set, and for the test set, divided in positives vs. negatives. (*b*) PCA reduction for GlanceNet over the its five latent factors. (*c*) PCA reduction for CBNM; the dotted line indicates the separating hyperplane predicted in the second phase. (*d*) Leakage % for CBNM, CG-VAE and GlanceNet.

For both competitors, we encode `shape` using a 3D one-hot encoding and `size` and `rotation` as continuous variables. During training, we use the `shape` annotation for conditioning the prior $p(\mathbf{Z} \mid \mathbf{Y})$ of the GlanceNet. The first two PCA components of the latent representations acquired by our GlanceNet and by a CBNM are shown, rotated so as to be separable on the first axis, in Fig. 5 (b, c): in both cases, it is possible to separate positives from negatives based on the obtained representations in the five latent dimensions. As shown in Fig. 5 (d), this means that both CBNM and CG-VAE suffer from very large leakage, $80\%$ and $98\%$, respectively. In contrast, OSR allows us to correctly identify and reject almost all test instances, leading to negligible leakage even in this disentangled setting.

# 6   Related Work

**Concept-based explainability.** Concepts lie at the heart of AI [56] and have recently resurfaced as a natural medium for communicating with human stakeholders [11]. In explainable AI, this was first exploited by approaches like TCAV [57], which extract local concept-based explanations from black-box models using concept-level supervision to define the target concepts. Post-hoc explanations, however, are notoriously unfaithful to the model's reasoning [58–60]. CBMs, including GlanceNets, avoid this issue by leveraging concept-like representations directly for computing their predictions. Existing CBMs model concepts using prototypes [2, 3, 16, 17] or other representations [1, 20, 4, 5], but they seek interpretability using heuristics, and the quality of concepts they acquire has been called into question [61, 19, 7, 8]. We show that disentangled representation learning helps in this regard.

**Disentanglement and interpretability.** Interpretability is one of the main driving factors behind the development of disentangled representation learning [62–64]. These approaches however make no distinction between interpretable and non-interpretable generative factors and generally focus on properties *of the world*, like independence between causal mechanisms [9] or invariances [43].

Interpretability, however, depends on human factors that are not well understood and therefore usually ignored [12, 65]. The link between disentanglement and interpretability has never been made explicit. Importantly, in contrast to alignment, disentanglement does not require that the map between matching generative and learned factors preserves semantics. We remark that other VAE-based classifiers either do not tackle disentanglement or are unconcerned with concept leakage [66, 67, 36].

**Disentanglement and CBMs.** Neither the literature on disentanglement nor the one on CBMs have attempted to formalize the notion of interpretability or to establish a proper link between the latter and disentanglement. The work of Kazhdan et al. [68] is the only one to compare techniques for disentangled representation learning and concept acquisition, however it makes no attempt at linking the two notions. Our work fills this gap.

## Acknowledgments and Disclosure of Funding

The research of ST and AP was partially supported by TAILOR, a project funded by EU Horizon 2020 research and innovation programme under GA No 952215.

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
