# A  Implementation details

## A.1  GlanceNet and CBNM Architectures

In all experiments, we used exactly the same architecture and number of latent variables for both GlanceNets and CBNMs to ensure a fair comparison.

**Encoder architectures:**

- *dSprites*: We chose a rather standard architecture [52]. It comprises six 2D convolutional layers of depth 32, 32, 64, 128, 256, and 256, respectively, all with a kernel of size 4, stride 2, and padding 1, and followed by ReLU activations. The output is flattened to a vector and passed through a dense layer to obtain the mean $\boldsymbol{\mu}(\mathbf{x})$ and (diagonal) variance $\boldsymbol{\sigma}(\mathbf{x})$ of the encoder distribution $\mathcal{N}(\mathbf{Z} \mid \boldsymbol{\mu}(\mathbf{x}), \operatorname{diag}(\boldsymbol{\sigma}(\mathbf{x})))$.
- *MPI3D*: We used the same architecture with slightly different convolutional depths of 32, 32, 64, 64, 128, and 256, changing also the kernel size to 3 and removing padding, as per [52].
- *CelebA*: We leveraged the architecture of Ghosh et al. [55], which is a common reference for VAE models on CelebA-64 [69]. The encoder is composed of four convolutions of depth 128, 256, 512, 1024 respectively, all with kernel size of 5, stride of 2, followed batch normalization and ReLU activation.

The models had exactly as many latent variables as generative factors for which supervision is available, which in our three data sets are 7, 21, and 10, respectively.

**Decoder architecture:** All models share the same decoder architecture, obtained by stacking:

- A 2D convolution on the latent space with a filter depth of 256, kernel size of 1, and stride of 2, followed by the ReLU activation;
- Five transposed 2D convolutions of depth 256, 256, 128, 128, 64, 64, and `num_channels`, respectively, all with kernel of size 4 and stride 2.

Here, `num_channels` is either 1 (dSprites) or 3 (MPI3D and CelebA). The shape of the last layer was chosen so as to match the dimension of the input image. Additional details can be found in the various Tables in this appendix.

## A.2  Supervision and Training

**Concept-level supervision.**  Depending on the supervision provided, only a fraction of the inputs was made available during training with their generative factors. In dSprites and MPI3D all generative factors are matched by the models, whereas in the case of CelebA we restricted learning to those 10 attributes that are best fit by the CBNMs, namely: `bald`, `black hair`, `brown hair`, `blonde hair`, `eyeglasses`, `gray hair`, `male`, `no beard`, `smiling`, and `wearing hat`. Both CBNMs and GlanceNets are jointly trained, meaning that optimization steps for the concepts and label supervision are taken simultaneously. Whenever concept supervision is lower than 100%, for those examples without concept annotations we trained both models using label supervision only. We did not evaluate other training strategies available for CBNMs (e.g., sequential training [20]) as these appear to bring no benefit in terms of either performance nor leakage.

**Optimization setup.**  In all experiments, we used the Adam optimizer [70] with default parameters $\beta_1 = 0.9$ and $\beta_2 = 0.999$. For dSprites, we used a batch size of 64 and fixed learning rate to $\eta = 4 \cdot 10^{-4}$, while for MPI3D and CelebA we used a batch size of 100 and annealed the learning rate from $10^{-7}$ to $\eta_{MPI} = 10^{-3}$ and $\eta_{CelebA} = 10^{-4}$, respectively. To prevent overfitting, in CelebA we multiplied the learning rate by a factor of 0.95 in each epoch and apply early stopping on the validation set, with a patience of 10 epochs.

Prior to training, we selected a reasonable value for the following hyper-parameters:

- $\beta$: the weight of the KL divergence in Eq. (4).
- $\gamma$: the weight of the loss on the generative factors in Eq. (5).
- $\lambda$: the weight of the cross-entropy loss over the label, which is left implicit in Eq. (4).

Table 1: Structure of the encoder network used for dSprites.

| INPUT SHAPE | LAYER TYPE | PARAMETERS | ACTIVATION |
|---|---|---|---|
| $(64, 64, 1)$ | Convolution | depth=32, kernel=4, stride=2, padding=1 | ReLU |
| $(32, 32, 32)$ | Convolution | depth=32, kernel=4, stride=2, padding=1 | ReLU |
| $(16, 16, 32)$ | Convolution | depth=64, kernel=4, stride=2, padding=1 | ReLU |
| $(8, 8, 64)$ | Convolution | depth=128, kernel=4, stride=2, padding=1 | ReLU |
| $(4, 4, 128)$ | Convolution | depth=256, kernel= 4, stride=2, padding=1 | ReLU |
| $(2, 2, 256)$ | Convolution | depth=256, kernel=4, stride=2, padding=1 | ReLU |
| $(1, 1, 256)$ | Flatten | | |
| $(1, 256)$ | Linear | dim=7+7, bias = True | |

Table 2: Structure of the encoder network used for MPI3D.

| INPUT SHAPE | LAYER TYPE | PARAMETERS | ACTIVATION |
|---|---|---|---|
| $(64, 64, 3)$ | Convolution | depth=32, kernel=3, stride=2 | ReLU |
| $(32, 32, 32)$ | Convolution | depth=32, kernel=3, stride=2 | ReLU |
| $(16, 16, 32)$ | Convolution | depth=64, kernel=3, stride=2 | ReLU |
| $(8, 8, 64)$ | Convolution | depth=64, kernel=3, stride=2 | ReLU |
| $(4, 4, 64)$ | Convolution | depth=128, kernel= 3, stride=2 | ReLU |
| $(2, 2, 128)$ | Convolution | depth=256, kernel=3, stride=2 | ReLU |
| $(1, 1, 256)$ | Flatten | | |
| $(1, 256)$ | Linear | dim=21+21, bias = True | |

Table 3: Structure of the encoder network used for CelebA.

| INPUT SHAPE | LAYER TYPE | PARAMETERS | FILTER | ACTIVATION |
|---|---|---|---|---|
| $(64, 64, 3)$ | Convolution | depth=128, kernel=5, stride=2 | BatchNorm | ReLU |
| $(30, 30, 128)$ | Convolution | depth=256, kernel=5, stride=2 | BatchNorm | ReLU |
| $(13, 13, 256)$ | Convolution | depth=512, kernel=5, stride=2 | BatchNorm | ReLU |
| $(5, 5, 512)$ | Convolution | depth=1028, kernel=5, stride=2 | BatchNorm | ReLU |
| $(1, 1, 1028)$ | Flatten | | | |
| $(1, 1028)$ | Linear | dim=10+10, bias = True | | |

Table 4: Structure of the decoder network.

| INPUT SHAPE | LAYER TYPE | PARAMETERS | ACTIVATION |
|---|---|---|---|
| $(\dim(\mathbf{z}))$ | Unsqueeze | | |
| $(\dim(\mathbf{z}), 1, 1)$ | Convolution | depth=256, kernel=1, stride=2 | ReLU |
| $(256, 1, 1)$ | Deconvolution | depth=256, kernel=4, stride=2 | ReLU |
| $(256, 2, 2)$ | Deconvolution | depth=128, kernel=4, stride=2 | ReLU |
| $(128, 6, 6)$ | Deconvolution | depth=128, kernel=4, stride=2 | ReLU |
| $(128, 14, 14)$ | Deconvolution | depth=64, kernel=4, stride=2 | ReLU |
| $(64, 30, 30)$ | Deconvolution | depth=64, kernel=4, stride=2 | ReLU |
| $(64, 62, 62)$ | Deconvolution | depth=`num_channels`, kernel=4, stride=2 | |

For dSprites, we found a good balance for $\lambda = \gamma = 100$, while for MPI3D we achieved good performance with $\lambda = 10^3$ and $\gamma = 7 \cdot 10^3$. We adopted the same hyperparameters choice for CelebA, with the exception that we reduced the reconstruction error by $0.01$. For all data sets, we cross-validated over different values of $\beta$ but we obtained better alignment performances with $\beta \approx 1$. This happens because we inject supervision on the latent factors (which is absent in regular $\beta$-VAEs [43]).

### A.3 Implementation of leakage tests

**MNIST.** For this dataset, we considered only Multi-Latyer Perceptrons instead of convolutions. Both the encoder and the decoder are composed by two linear layers with depth 128, and a dense layer connected to the latent space and to the input space, respectively. Further details are on Table 5.

For the GlanceNet we considered a latent space of dimension 10 where the supervision on the 4 and 5 digits is used to fit the $\{z_4, z_5\}$ latent factors. These two, constitute the latent subspace where leakage occurs, while the other are useful only for reconstruction. Conversely, for the CBNM we considered only two latent factors.

During training of the latent encodings, we used stochastic gradient descent with learning rate $\eta = 0.001$, reducing it by $0.95$ in each epoch for both CBNMs and GlanceNets. The training was performed only on the 4 and 5 digits (in the usual training set partition for MNIST), for almost 50 epochs. Afterwards, we considered the open-set representations, restricted to $\{z_4, z_5\}$, as inputs for training a logistic regression for parity recognition. During the training, only the digits in the MNIST training set partition (exception made for 4 and 5) are considered, while performance are calculated on the test set.

**dSprites.** We adopted the same architecture in the upper section, except that we reduced the latent space to 5 dimensions. As a reminder, during training all sprites are almost fixed at the center, therefore additional factors of variations for its position are needless. The training was performed over 300 epochs for both GlanceNets and CBNMs, with $\eta = 4 \cdot 10^{-4}$. After training, the representations of the open-set sprites (in which position is no longer fixed) are used to fit a logistic regression. In this case, the labels depend on whether the sprite is located at bottom-left corner or at the upper-right one, for more information refer to Fig. 5.

The classification performance was evaluated on a held-out test set for both models, under an 80/20 train/test split.

Table 5: Encoder and Decoder structures for MNIST

| TYPE | INPUT SHAPE | LAYER TYPE | PARAMETERS | ACTIVATION |
|---|---|---|---|---|
| ENCODER | | | | |
| | $(28, 28)$ | Flatten | | |
| | $(784)$ | Linear | dim=128, bias=True | ReLU |
| | $(128)$ | Linear | dim=128, bias=True | ReLU |
| | $(128)$ | Linear | dim=10+10, bias=True | |
| DECODER | | | | |
| | $(\dim(\mathbf{z}))$ | Linear | dim=128, bias=True | ReLU |
| | $(128)$ | Linear | dim=128, bias=True | ReLU |
| | $(128)$ | Linear | dim=728, bias=True | |
| | $(728)$ | Unsqueeze | | |

## B  DCI framework

In our case study, we are interested into DCI in [29]) maps that linearly connect the $\mathbf{z}'s$ to the $\mathbf{g}'s$. In order to evaluate alignment performances, the inverse map $\alpha^{-1} : \mathbb{R}^k \to \mathbb{R}^{n_I}$ is constructed from the latent space to the span of the $n_I$ generative factors. The latent representations and generative factors were normalized in the $[0, 1]$ interval prior to learning.

## B.1 Alignment and explicitness

The importance weights of this map are the absolute-values of the weights in the linear matrix of $\alpha^{-1}$, indicated as $B \in \mathbb{R}^{k \times n_I}$ in the main text. Then, the importance weights are used to evaluate the dispersion of the learned weights. To this end, we measure each Shannon entropy $H_j$ on all $k$ latent factors:

$$H_j = -\sum_{i \in 1}^{n_I} \bar{b}_{ji} \log_n \bar{b}_{ji} \quad \text{where} \quad \bar{b}_{ji} = b_{ji} \Big/ \sum_{\ell=1}^{n_I} b_{j\ell} \tag{6}$$

Then, the average alignment is calculated as:

$$\text{alignment} = 1 - \sum_{j=1}^{k} \rho_j H_j \quad \text{where} \quad \rho_j = \sum_{i=1}^{n_I} b_{ji} \Big/ \sum_{j'=1,i=1}^{k,n_I} b_{j'i} \tag{7}$$

and ranges in $[0, 1]$. Similarly, the quantity:

$$\tilde{b}_{ji} = b_{ji} \Big/ \sum_{\ell=1}^{k} b_{\ell i} \quad \text{and} \quad \tilde{H}_i = \sum_{j=1}^{k} \tilde{b}_{ji} \log_k \tilde{b}_{ji} \tag{8}$$

is the *completeness* of the latent representation, a measure akin to alignment (Eq. (7)) that quantities the degree to which each generative factor correlates with *distinct* latent factors. Alignment and completeness relate to different properties of the map: the higher the *alignment*, the more each $Z_j$ depends on variations of only a single $G_i$. On the other hand, learning multiple $Z_j$'s capturing a single $G_i$ reduces the *completeness*. As an illustrative example, consider the matrix:

$$B = \begin{pmatrix} 1 & 0 & 0 \\ 0 & 0 & 1 \\ 0 & 1 & 0 \\ 0.2 & 0 & 0 \\ 0 & 0.2 & 0 \end{pmatrix}$$

From the above definitions, one gets $alignment = 1$ and $completeness < 1$. This follows since each Shannon entropy for the alignment score is zero (as it is related to the rows), whereas the Shannon entropy for the completeness is greater than zero (it refers to the columns). Moreover, each latent variable $z_i$ depends only on the variations of a single generative factor $g_j$.

We also calculate the explicitness of the map $\alpha$, which is related to the mean squared error (MSE) of the prediction. Since the MSE for random guessing for a variable in the $[0, 1]$ interval is equal to 1/6, the explicitness becomes:

$$\text{explicitness} = 1 - 6 \cdot \text{MSE}$$

## B.2 Empirical evaluation

For dSprites and MPI3D, all DCI quantities were calculated with the built-in evaluation code provided by `disentanglement_lib`, [31]. For CelebA, since the 40 attributes in CelebA are not exhaustive for the image generation, we implemented computed DCI as follows: *(i)* we first converted the $J$ attributes $\mathbf{z}_J$ and $\mathbf{g}_J$ connected to `hair type` to a single concept $h$ and fit the model with Lasso regression to predict $g_h$ from $\mathbf{z}$. Then, *(ii)* we trained a Logistic Regression with $l1$ penalty to predict the remaining $\mathbf{g}'s$. Finally, we took both weights in *(i)* and in *(ii)* to compute the matrix $B \in \mathbb{R}^{6 \times 6}$. In this way, we determined alignment and explicitness for CelebA. We chose the lasso coefficient $\lambda = 0.01$ for both regressions.

## C Open-Set Recognition Mechanism

In this section, we provide additional details on the OSR mechanism introduced in Section 3. Our method adapts the one of Sun et al. [36], which distinguishes between closed-set and open-set data points by combining a reconstruction check $\Gamma_r$ with a localization check $\Gamma_{ls}$. The overall OSR check is given by:

$$\hat{T} = \Gamma_r \wedge \Gamma_{ls} \tag{9}$$

After completing the training process, all the training instances are passed to the model to evaluate the thresholds:

- The reconstruction threshold $\eta_r$ is the maximum real number such that a fixed percentage of training examples have reconstruction error less or equal to it. At test time, given an instance $\mathbf{x}$, let $\hat{\eta} = \|\mathbf{x} - \hat{\mathbf{x}}\|^2$ be the reconstruction error. Then, $\Gamma_r = 1$ (i.e., the check passes) if the empirical reconstruction error is less than the threshold, $\hat{\eta} < \eta_r$, otherwise $\Gamma_r = 0$.

- The latent-space distance thresholds are evaluated for each class-prototype embedded in the latent space $\mu_y = \mathbb{E}_{p(\mathbf{z}|y)}[\mathbf{z}]$. For each of them, we first evaluated the relative distance between point belonging to the class $y$ and the prototype $\mu_y$. Then, we evaluated a threshold $\eta_y$ on the distances, as to include a fixed percentage of training instances into the set $\mathcal{B}_y = \{\mathbf{z} : \|\mu_y - \mathbf{z}\| < \eta_y\}$. At test time, those points that do not belong to any set $\mathcal{B}_y$ are predicted as open-set instances, i.e. $\Gamma_{ls} = 0$, otherwise $\Gamma_{ls} = 1$.

In our experiments, the threshold are obtained by fixing both reconstruction and latent space distance to keep the $95\%$ of training data. In the case of $\eta_y$, this quota has been reached singularly for each $\mathcal{B}_y$, thus obtaining different values $\eta_y$'s from one another. Finally, combining both rejection methods we are sure the model would predict as closed-set at least the $90\%$ of training instances.

## D  Concept Leakage in MNIST

We report here additional details for the concept leakage test on MNIST, which has been originally introduced by Margeloiu et al. [8]. The experiment has two stages:

1. At train time, the model is trained to align its representations to the concepts of $4$ and $5$, by passing full supervision on them. Both CBNMs and GlanceNets are allotted two latent concepts, which we denote $(Z_4, Z_5)$. There is no downstream classification task in this stage.

2. At test time, all MNIST images, excluding those of $4$'s and $5$'s, are encoded using the learned encoder and used to learn a classifier of even vs. odd digits. The performance of the resulting classifier, applied to non-$\{4, 5\}$ images, is then computed.

In this experiment, concept leakage occurs if the accuracy on the downstream task is above the $50\%$.

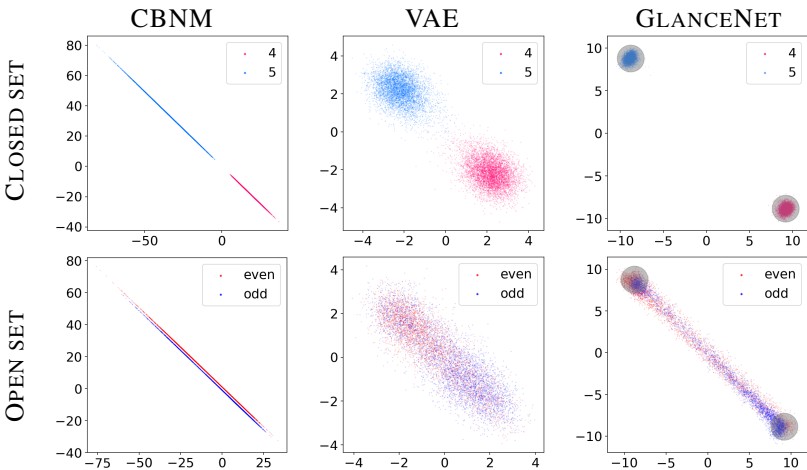

Figure 6: **Latent space representation for MNIST.** On the first row, we report the representations for $4$ and $5$ as fitted by CBNM, VAE and GlanceNet, respectively. On the second row, we display the scattering plot for points only belonging to the open set. For CBNM, we separated even and odd instances by $\Delta y = 2$, since their representations strongly overlap. All plots comprise only the $z_4, z_5$ axes.

### D.1  Qualitative results

In Fig. 6, we show the latent space representations for different models on the MNIST leakage test, for both closed-set and open-set data points. To illustrate the contribution of our mixture prior,

in addition to the CBNM and GlanceNet models, we also considered a simpler supervised VAE model. This model has the same encoder, decoder, and classifier as the GlanceNet, but uses a regular Gaussian prior[4]. We found that this model achieved a similar level of leakage to CG-VAE. We display in Fig. 7 the reconstruction of a few random examples output by GlanceNet: the reconstructions of all instances belonging to the open-set greatly deviate from the original.

8 9 0 1 2 3 4 5    5 4 5 4 5 4 4 5
6 7 8 0 1 2 3 4    5 4 5 5 5 4 5 4
7 8 9 7 8 6 4 1    4 5 4 4 5 5 4 4

Figure 7: **MNIST reconstruction with GlanceNet.** On the left we reported the original digits, whereas on the right the reconstruction with the learned decoder. All images have been inverted in the black and white scale.

# E   Concept Leakage on dSprites

In this section, we report additional material for the dSprites concept leakage experiment. This experiment resembles the previous one on MNIST:

1. At training time, a CBM learns the representations of `shape`, `size` and `rotation` by receiving supervision on all possible variations of these factors. On the other hand, no variation of factors `pos_x` and `pos_y` are observed, in the sense that the position of the training sprites is fixed to the center of the image. We fit the CBNM and the GlanceNet with 5 latent factors to learn the representations. Again, no downstream classification task appears at this stage.

2. At test time, the encoder is kept fixed for different variations varying of the factors `pos_x` and `pos_y` are observed. The downstream task in this phase amounts to recognizing whether a sprite lies in top-right or in the bottom-left corner of the image.

In this experiment, concept leakage occurs if the accuracy on the downstream task is above the 50%.

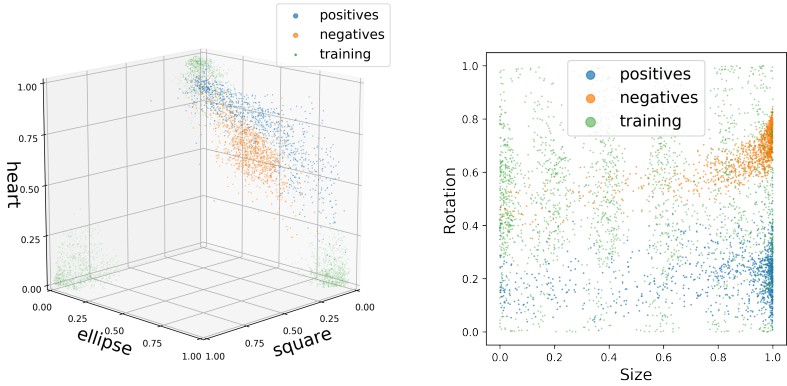

Figure 8: **Concept space representation of GlanceNet for dSprites.** On the left, we show the projections on the one-hot encoded `shape` subspace, whereas on the right we project on the {`size`, `rotation`} subspace. We include the representations for training points, positive and negative ones.

## E.1   Qualitative results

We also include qualitative results for GlanceNet and on dSprites for closed set and open set data points. In Fig. 8 we display the projections of train and test points on the two different latent subspaces

---

[4]For the VAE model, we chose the Gaussian prior in [34], i.e., $p(\mathbf{z}) = \mathcal{N}(\mathbf{z}|0, 1)$.

(see caption). In both of them, positives and negatives representations are well separated from each other, implying substantia leakage. We also evaluated the reconstruction quality during training and testing and reported some of them in Fig. 9. Notably, almost all points are recognized to be open set instances thanks to the reconstruction threshold.

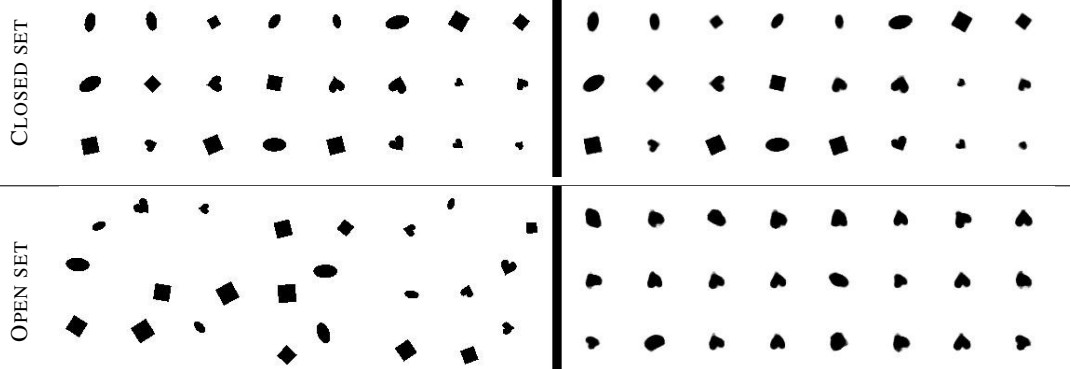

Figure 9: **Reconstruction for dSprites on train and test with GlanceNet.** On the upper panel, we report the reconstructions of the sprites belonging to the closed set. On the lower one, the reconstructions of the open set points. Like MNIST, all images have been inverted in the black and white scale.

## F    Additional results for GlanceNets and CBNMs in CelebA

In this section, we discuss additional results for CBNMs vs. GlanceNets on the CelebA dataset. We first report the accuracy of the learned concepts on the supervised latent factors for both CBNMs and GlanceNets in CelebA. Then, we examine two variants of GlanceNets varying the dimension of the unsupervised factors in the latent space: a $\beta$-VAE with 20 latent factors and a $\beta$-TCVAE with 40 latent factors, [39]. This variant includes an additional loss term given by the Total Correlation (TC) of the model posterior $q_\phi(\mathbf{z}) = \mathbb{E}_{p_\mathcal{D}(\mathbf{x})}[q_\phi(\mathbf{z}|\mathbf{x})]$:

$$(\beta - 1) \cdot \mathsf{KL}\big(q_\phi(\mathbf{z})|| \prod_{i=1}^{k} q_\phi(z_i)\big) \tag{10}$$

where $\beta$ denotes the strength hyper-parameter. Both the $\beta$-VAE and the $\beta$-TCVAE receive supervision only on the 10 generative factors that are fitted in the CBNM. A the end of the section, we report traversals for the models with 40 latent factors.

### F.1    Concepts Accuracy

We report the concepts accuracy for both CBNMs and GlanceNets in Fig. 10, with 10 latent dimensions and the TCVAE variant. The difference in concept accuracy between GlanceNet (both variants) and CBNMs is negligible, with GlanceNets showing slightly higher variance when the percentage of concepts supervision is very small. This highlights how, in terms of concept accuracy, the two classes of models are essentially indistinguishable, even though they are in terms of alignment.

### F.2    Performances upon variations of the latent space dimension

Here, we show the behavior of the metrics upon increasing the dimension of the latent space. The first variant of GlanceNets, based on $\beta$-VAE, was fitted with $\beta \approx 1$, with a latent space of dimension 20. The second variant is a TCVAE, trained with a weight of the total correlation $\beta = 10$ for all concepts supervisions, exception made for the 100% run, where we found better results with $\beta = 0.5$. We measured alignment and explicitness for both variants of GlanceNets by restricting on only those 10 latent factors where supervision were provided. This is in line with the notion of alignment, since we are interested in measuring the interpretability of the model, not the disentanglment among different components.

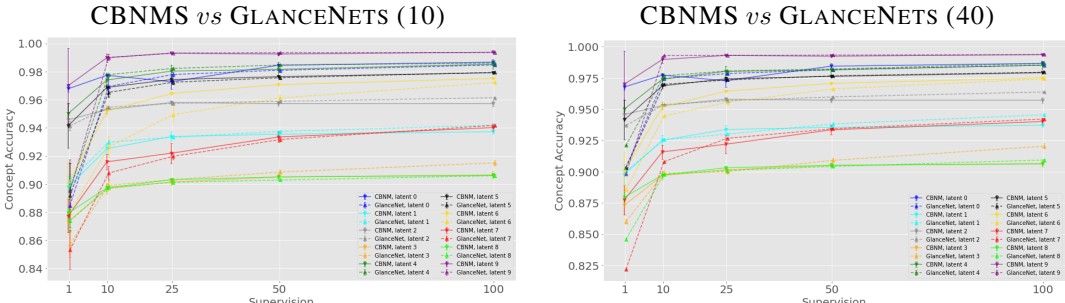

Figure 10: Concepts accuracy for CBNMs *vs* GlanceNets. Different colors refer to the distinct attributes for which supervision is provided. The solid line is reserved to CBNMs, whether GlanceNets are displayed with a dotted line. On the left, CBNMs vs GlanceNets with a latent space of dimension 10. On the right, CBNMs vs GlanceNets with TCVAE variant and a latent space of 40.

In Fig. 11 we report the results obtained, including the original variant with 10 latent dimensions. For the $\beta$-VAE (20) and TCVAE (40) we can see the improvement provided by extending the latent space. The latter achieves particular high values of alignment.

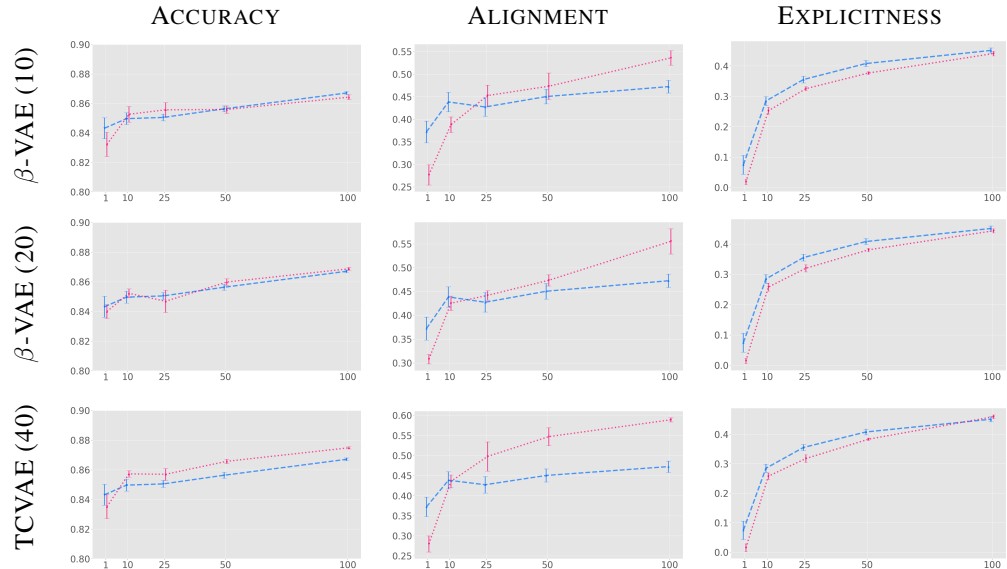

Figure 11: Accuracy, Alignment and Explicitness metrics for CBNMs *vs* GlanceNets. For each row we vary the comparison with variants of GlanceNets: $\beta$-VAE (10) refers to the model we reported in the main text, $\beta$-VAE (20) is a variant with 20 latent dimensions, and TCVAE (40) is the model based on a TCVAE with 40 latent dimensions.

### F.3 Latent traversals

We finally report in the traversals for some of the supervised attributes, obtained by the GlanceNet TCVAE with full supervision on the concepts. We excluded the traversals of the attributes HAT and BALD since the generator failed to reproduce them faithfully. The others are well captured by the model, as we reported in Fig. 12.

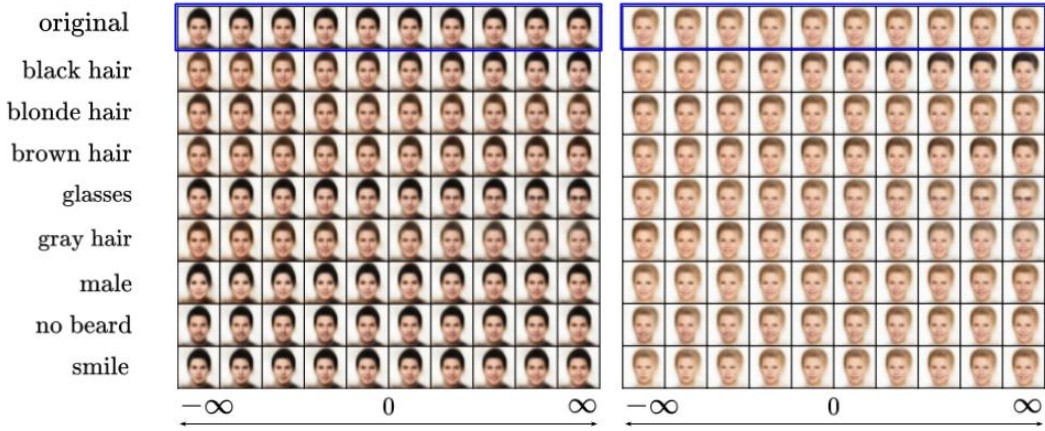

Figure 12: Latent traversals on two test images. In each row, we report the result of changing a single latent factor $Z_i$ (from $-5$ to $+5$) while keeping fixed the others.