# OpenReview forum: "GlanceNets: Interpretable, Leak-proof Concept-based Models"
_NeurIPS.cc/2022/Conference — NeurIPS 2022 Accept_

### Official Review · Reviewer_mGx1 · 2022-07-10

**Rating:** 5
**Confidence:** 4
**Soundness:** 2 fair
**Presentation:** 2 fair
**Contribution:** 2 fair

**Summary:**

For improving the interpretability of concepts in concept-based models (CBMs), in this paper, the author(s) proposed a model, GlanceNet, to utilize the techniques from disentangled representation learning and open-set recognition for achieving alignment and avoiding concept leakage. As the author(s) claimed that experiments on several datasets showed the advantages of the proposed method.

**Questions:**

As for the Weaknesses mentioned above, to be specific, I have the following questions/comments which need further explanation from the author(s). Please correct me if I am wrong. Thanks.

1. Could you please provide more details about threshold $\eta_y$? In Line 214, the author(s) mentioned that "...choosing the thresholds as to include the 95% of training examples to work well in our experiments.", what is the effect of different percentage samples? To include the 95% of training examples, for different $y$, did you use different values? Or the same value? Did you choose $\eta_y$ based on the results on the test set?

2. Figure 3, did different data partitions (using different random seeds) have different effects on the results? In Checklist, for Question 3 (c), the author(s) answered "the error bars for CelebA was reported...".  For CelebA, taking the 100% sample as an example, the maximum fluctuation reached about 0.1. For MPI3D, the difference between curves is obvious in visual effect, but the actual difference was also about 0.1. For the calculation on MPI3D, is the average based on multiple data partitions? Or is it just one partition? If the latter, is it possible that the method in this paper is not so superior?

---
---

In addition, some other tiny issues/typos:

(1) Please give the full name of DCI when it first appears;

(2) When using mathematical equations, sometimes there are punctuation marks and sometimes there are not. Please use a consistent format;

(3) Please check and use "em dash" carefully in the paper;

(4) The numbers on the vertical axis in Figure 4 (c) are too small;

(5) What does $\mathbf{s(x)}$ mean in Eq.(1)? What is difference of  $\mathbf{Z}_J$ and $\mathbf{z}_J$ in lines 141-142;

(6) There are many formatting inconsistencies in the cited references. Please check and correct them carefully.








**Limitations:**

Yes.

**Strengths And Weaknesses:**

This paper is generally well written, I have the following comments:

### Strengths
---

1. Defining interpretability in terms of alignment between the model’s representation and an underlying data generation process;

2. Avoiding concept leakage by utilizing the viewpoint from OOD;

3. Empirical experiments were performed to show the superiority of the proposed method.

---
---

Although the discussion and analysis in this paper are interesting, I think this paper has the following deficiencies or unclear points (Please see the "Questions" section below for details).

### Weaknesses
---

1. Some experimental designs are not very clear;

2. Some experimental results need further clarification.

---

---

> ### Author Response · Authors · 2022-08-02
> **Reply to Reviewer mGx1**
>
> Thank you for the thoughtful and detailed review.  We appreciate that you found the paper well written and our contributions significant.  Below you can find replies to specific points raised in your review.  We will soon update the paper to take care of all the minor mistakes and typos you reported.
>
> **Choice of $η_y$**: The procedure is: (1) We trained a GlanceNet on the training set, which gives us a centroid for each class $y$, encoded in the Gaussian Mixture prior; (2) We computed distances from the centroid to each training (not test) example of the same class; (3) We chose the distance threshold $η_y$ so that it covers 95% of the training (not test) examples belonging to that class.  Each class has its own threshold.  We did not explore different percentages, as 95% works very well in our experiments and it intuitively accounts for a small percentage of outliers.  Notice that [Sun2020] also uses a heuristic for choosing the threshold.  We added a new Section C on the OSR settings to the appendix where you can find more details.
>
> **Missing error bars in dSprites and MPI3D and significance of our results**:  We did report error bars on CelebA as this data set is much harder than the others: it is not as large (120k images compared to 700k for dSprites and 1m for MPI3D; the test set is 10% of the total in both cases) and - critically - it does not contain all possible combinations of generative factors.  The other data sets, on the contrary, are very large and exhaustive.  Preliminary evidence suggests that performance is stable across methods on dSprites and MPI3D.  Nonetheless, we are running additional experiments and will update the paper and the rebuttal as soon as we have more information.
>
> Based on these results, we believe that our message - that is, that GlanceNets achieve better alignment and avoid leakage while reaching the same prediction performance as CBMs and similar baselines - has enough empirical evidence in support of it.  Indeed, our results show that:
>  - The message does hold on two big data sets (70k test examples for dSprites and 100k for MPI3D).
>  - The message holds also for CelebA, a much harder, real-world (non-exhaustive) data set, for which we report confidence information over seven random seeds.
>
> Moreover, GlanceNets can be built on top of any state-of-the-art disentangled VAE architecture.  We hypothesize that alignment (and explicitness) could be improved this way as well as by integrating more unsupervised uninterpretable latent variables.  To verify this, we included an additional experiment on CelebA by integrating TCVAEs and a bigger latent space.  These new results show that:
>  - The message holds also on CelebA, using TCVAE-based GlanceNets, and we report confidence information over five random seeds.  In this new set of experiments, GlanceNets obtain even better alignment on CelebA, see Appendix F.
> We believe this provides quite some evidence that our claims are well founded.
>
> **DCI is undefined**:  Well spotted.  DCI stands for Disentanglement Completeness and Informativeness, which is used as an acronym in the paper.  We updated the paper accordingly.
>
> **Meaning of $s(x)$**: As we mentioned in line 48, these are the per-class scores (or logits) that are passed to the softmax to predict p(y|z).
>
> **Difference between $z$ and $Z$**.  As is common in probability theory, we use upper case $Z$ to refer to random variables, lower case $z$ to denote constants, and bold to denote ordered sets of random variables or constants.  We will briefly outline our notation in the paper in Section 2.

---

> > ### Author Response · Authors · 2022-08-06
> > **Comment to reviewer mGx1**
> >
> > Dear Reviewer,
> >
> > We would like to follow up to see if our response addresses your concerns or if you have further questions. We would really appreciate the opportunity to discuss this further and know whether our response has already addressed your concerns. Thank you again!

---

### Official Review · Reviewer_Xfpf · 2022-07-10

**Rating:** 7
**Confidence:** 4
**Soundness:** 3 good
**Presentation:** 4 excellent
**Contribution:** 4 excellent

**Summary:**

This paper introduces GlanceNets, a novel family of concept bottleneck models that leverages disentanglement learning and open-set recognition to avoid information leakage and generate meaningful concept-based explanations for predictions. The authors motivate GlanceNet by first providing an explicit definition of interpretability in terms of concept alignment to interpretable data generative factors, establishing a connection between disengagement learning and concept learning. Then, they use this definition to define an autoencoder-based architecture that, through full or weak concept supervision, learns to extract a latent code that aligns with interpretable factors of variation (i.e., concepts) and uses this code for predicting a downstream label. With three sets of experiments, this paper shows that GlanceNets are able to maintain high task accuracy while avoiding unnecessary leakage in their learned concept encodings.

**Questions:**

After reading the paper a few times, I am still unclear about the following items:

1. One thing that is particularly missing in the experiments is an evaluation of how well GlanceNets can predict concepts (i.e., the accuracy of using $p(g_i | x) = \sigma(z_i)$ to predict the $i$-th interpretable generative factor). How does this accuracy compare to that of a vanilla concept bottleneck model? The alignment may implicitly capture this in some way, however, in practice, this is an important metric as it determines how accurate its explanations are.
2. In the context of experiment 5.1, it is unclear what entails to train a vanilla Concept Bottleneck Model with only a percentage of its samples with concept annotations. Given that these models require full-concept supervision, can you please clarify what was the procedure used to train them if only a portion of its training data has concept annotations? Did you ignore the concept-prediction loss term for samples that do not have annotations in them or did you simply reduce the training set size to be that of samples that have annotations?
3. How are the concept bottleneck models used in the paper's evaluation trained? Are they trained jointly, sequentially, or independently? I might’ve missed something but I was unable to find an answer to this in the paper and its supplementary material.
4. My biggest concern with the evaluation of this method is that some important details of the setup for the experiments in section 5.2 are a bit unclear. For example, when training the different baselines in the MNIST task, it is unclear what the downstream task (i.e., $y$) is. From the supplementary material, it is my understanding that parity prediction is not the downstream task used to learn the concept latent codes for both CBNM and GlanceNet. Is the downstream task therefore just predicting the image’s number? This should be clarified as it is a key component in understanding what sort of supervision these encodings get when being learned. Similarly, for the dSprites experiment, what is the downstream task used to train the different baselines? From Figure 5. (a), it is unclear how samples in the square corresponding to the training set are labeled, and thus it is unclear what sort of training supervision the concept encodings get during training. Clarifying these things would certainly improve this evaluation substantially.

In terms of suggestions for the presentation, the paper is extremely well-written and clear (thanks for this!). Nevertheless, the following typos may be worth addressing for a final version:

1. There is a typo in the title: “Interpretabile” should be “Interpretable”
2. Traditionally, the acronym “CBM” is used for “Concept Bottleneck Model” rather than “Concept-based Model”. To avoid confusion, I would suggest following this convention and using CBM for “Concept Bottleneck Model” (rather then “CBNM”) and something else for “Concept-based Model” (e.g., “CM” or “CLM” for “Concept learning model” as in Mahinpei et al.).
3. Line 193: the citation for [37] is a bit awkwardly placed.
4. Figure 3: it would be helpful if the axes are labeled in these plots.
5. Line 285: “Eastwood and Williams [29]” seems to be misplaced. Maybe it is intended to be just “[29]”?
6. Line 309: “trained” should be “training”.
7. Figure 4:  it would be helpful if the axes are labeled in these plots.
8. Line 330: “Fig. 4 (a)” should probably be “Fig. 5 (a)”.

**Limitations:**

The authors do a good just at discussing some limitations of their work (e.g., how their work assumes that all interpretable generative factors are independent of each other). As for potential negative societal impacts, I was unable to find a discussion on this topic in the paper or in the checklist. Nevertheless, I do not see any immediate potential negative societal impacts from this work.

**Strengths And Weaknesses:**

Thank you for the very insightful and interesting work presented in this paper. After carefully reading your paper and its supplementary material I believe that its main strengths are:

1. The method is novel and it addresses an important gap in the literature by being, to the best of my knowledge, the first work to design an architecture that explicitly addresses leakage in CBMs.
2. This work provides a clear and insightful link between disentanglement learning and concept learning.
3. The evaluation of the proposed methods is extensive and well-motivated.
4. The work is clearly placed within its literature and the paper offers an extensive amount of references and resources for the interested reader.
5. The paper is very well-written and its presentation is very clear.

I found the following things to be potential weaknesses of this paper:

1. As opposed to conventional practice, error bars are not included in all results (e.g., dSprites and MPI3D plots in Figure 3).
2. A few crucial design elements of the experiments are unclear (see below).
3. The code provided as part of the Supplementary material is lacking clear instructions and/or annotations for one to know how to follow it.

---

> ### Author Response · Authors · 2022-08-02
> **Reply to the Reviewer Xfpf**
>
> Thank you for reviewing our paper, it is really encouraging to see that you appreciated the novelty and significance of our contribution.  We hope that the changes we propose below and the additional clarifications will improve the paper even further.
>
> **Missing error bars in dSprites and MPI3D**:  We did report error bars on CelebA as this data set is much harder than the competitors: it is not as large (120k images compared to 700k for dSprites and 1m for MPI3D; the test set is 10% of the total in both cases) and - critically - it does not contain all possible combinations of generative factors.  The other data sets, on the contrary, are very large and exhaustive.  Preliminary evidence suggests that performance is stable across methods on dSprites and MPI3D.  Nonetheless, we are running additional experiments and will update the paper and the rebuttal as soon as we have more information.
>
> **Code lacks clear instructions**. Thank you for pointing this out!  Admittedly, we did not have time to fully document the code.  We made an effort to update it in the supplementary and have included a simple README file.  We plan to clean it up much further in the near future.  We also intend to provide full support for the GitHub version.
>
> **Accuracy of learned concepts** This is a valid point.  We have evaluated the accuracy of the 10 interpretable concepts we use in CelebA for different models and amounts of supervision. These new results can be found in Section F. The plots show that (1) Accuracy is generally good for all models and concepts, (2) The difference in accuracy between GlanceNets and CBNMs is small. Paired with the fact that the alignment of GlanceNets is better than that of CBNMs, (1) and (2) highlight that concept accuracy itself is not enough to evaluate interpretability, as it does not consider correlations between concepts (which are high in CBNMs).  In contrast, our alignment measure does.
>
> **CBNM training**. We train the CBNM encoder and classifier jointly.  We did not take into consideration sequential or independent training of CBNMs because GlanceNets are also trained jointly.
>
> **Concept annotations** We added this information to Section A. The data sets we use contain full concept supervision for all interpretable factors (5 in dSprites, 7 in MPI3D, and 10 for CelebA).  In our experiments, we change the percentage of examples (from 1% to 100%) for which we show concept annotations to the model.  For both CBNMs and GlanceNets, the concept-level cross-entropy runs over the annotated examples only.
>
> **Downstream task for MNIST leakage experiment**. The experiment is set up as suggested in [Mahinpei21] and discussed in Section 2 (lines 79-91):  first, we learn a GlanceNet or CBNM encoder (and decoder) using concept supervision about what examples are 4’s or 5’s but no label supervision.  So there is no downstream task in this step.  In a second step, we train a classifier on top of the concepts learned in the previous step, using even vs odd labels.  We reported this information in Section D.
>
> **Construction of the dSprites leakage experiment**. When training the encoder (and decoder), the model observes training images where shape, size, and rotation take all possible values (and for which supervision is provided), but the sprite is fixed at position (0.5, 0.5).  This is represented by the green square in Fig 5(a).  In this step, there is no downstream task.  In a second step, a classifier is trained to distinguish between sprites (again, with all possible shapes, sizes, and rotations) that lie in the bottom left or top right areas of the space, represented by the blue and orange boxes in Fig 5(a).
>
> We acknowledge also typos you mentioned and we will include corrections in the revised version.

---

> > ### Author Response · Authors · 2022-08-06
> > **Comment to Reviewer Xfpf**
> >
> > Dear Reviewer,
> >
> > We would like to follow up to see if our response addresses your concerns or if you have further questions. We would really appreciate the opportunity to discuss this further and know whether our response has already addressed your concerns. Thank you again!

---

> > > ### Comment · Reviewer_Xfpf · 2022-08-06
> > > **Reply to Rebuttals and Updated Submission**
> > >
> > > Dear Authors,
> > >
> > > I would like to begin by thanking you for taking the time to go over our reviews and address some of our concerns. As mentioned in my original review, I find this work exciting and think there is good potential in this submission. I find the rebuttal's replies, and the updated submission, to rest any questions I had and to be aligned with the acceptance score I originally assigned to this paper. I wish you the best of luck with this submission and once more thank you for taking the time to reply to my questions!

---

### Official Review · Reviewer_jG58 · 2022-07-11

**Rating:** 6
**Confidence:** 4
**Soundness:** 3 good
**Presentation:** 3 good
**Contribution:** 2 fair

**Summary:**


This work introduces a new autoencoder architecture GlanceNet and proposes to advance the interpretability of concept-based models (CBMs) by introducing a definition of interpretability that is based on a supervised aligning of learned latent concepts with the factors of the underlying data generation, some of which maybe semantically interpretable to a user.

The latent space Z induced be the encoder q_theta(Z | X) is divided into two parts, Zj and Zbar_j.  Zj is for learning of k concepts which are used for task level classification p(Y | Zj) and for aligning with supervised generative factors G, which is measured via a DCI metric from the disentanglement literature.  Zbar_j is used by the decoder to reconstruct x_hat which is in turn novelly used by an open set recognition module to learn class prototypes that determine if a test set example follows the training distribution ( T=1 ) or not ( T=0 ) and can be used for rejecting test cases.  Additionally, a prior p(z | y) of concept prototypes is learned as a mixture of gaussians (one component per class) and used to encourage disentangled representation learning via penalizing the KL divergence between it and the latent space, KL( p(z|y) || q( z | x) ).

Over three image tasks with varying amounts of supervision for generative factors (1 to 100% of examples), and under the assumption that interpretable generative factors are disentangled from each other, the authors show the architecture performs comparably or better in terms of accuracy, and better in terms of the two interpretable metics, alignment and ``explictness” which measures how well the linear regressor employed by DCI fits the generative factors using Zj.

**Questions:**

1) Its unclear what the method does when T predicts a test example is out of domain.  Is it just not used in the results in section 5.1?
In 5.2 you show OSR helps with concept leakage, by rejecting 85% or almost all cases and then guessing random, which by definition will do get you better results for a binary task, but its not clear how OSR is used in 5.1 ( are samples rejected there too?).

2) What is the size of Zbar_j in these experiments?  I might be missing something but I didn’t see it in the appendix ?

3) How does the accuracy of your experiments compare with other non-concept based models in general?

**Limitations:**

Yes

**Strengths And Weaknesses:**

## Strengths:
The paper is novel for its proposed architecture, tying the definition of interpretability via the alignment of learned concepts to generative factors, and its utilitization of OSR for use in preventing concept leakage.  It additionally is shown to be quite performative against the concept whitening paper for synthetic cases ( first two experiments dSprites and MPI3D) & the third case CelebA which is semi-synthetic because the supervised factors themselves are used to cluster and create classes for the data (so y is conditioned on z which is supervised by G ).  The writing overall is fairly clear, though the use of T in the experiments for 5.1 was unclear to me and section 5.2 as a whole could be improved in terms of clarity.  The exact setup for learning the prototypes in the OSR module could have been made a little clearer.

## Weaknesses:
The authors need to make a more compelling case of if alignment to factors in this setting really leads to better interpretability because they make lots of assumptions to simplify the problem including
1) having generative factor supervision in the first place which seems unrealistic and limits its applicability, and
2) that there is no mixing of factors to create concepts since empirically they only show one to one factor -> concept mappings.  The alignment metric DCI gives higher value for fewer concepts features being used as predictors for generative factors, but even without allowing mixing, its quite possible the same factor could be transformed in many monotonic ways to create many concepts ( which is valid according to your setting ) and the metric in that case would be shown to be artificially low.
This latter assumption affects both the alignment and explictness metrics, and the latter is seems largely just a transform of the former after seeing its definition in the appendix and given the simplified setting.

Without human studies as to the utility of these explanations, its unclear how useful they are in this restricted setting, though I imagine they should be given the data is setup to be explained by the generative factors used.
More importantly for human studies of interpretability is how this work applies in general settings ( where generative factors aren’t given ) though this could be outside of the scope of this paper.

Overall its an interesting idea, but the paper could be made stronger by human evaluation of whether alignment in a simplified conservative setting is more “interpretable” compared with other CBMS.  Having an experiment which is less synthetic meaning where the classes aren’t explicitly based on the supervised generative factors given could help show this work would generalize to more real world settings and increase the significance of the findings.  If not, motivation and examples for how and where this setting, where generative factors with one to one mappings to concepts are given, are impactful is very much needed to highlight the significance and applicability of this work.

---

> ### Author Response · Authors · 2022-08-02
> **Reply to the Reviewer jG58**
>
> Thank you for the time and effort you spent reviewing our paper.  We will take care of all minor issues that you pointed out in a revised version of the manuscript.  Please find detailed replies below.
>
> **Concept Whitening**. Please note that we compare against Concept-bottleneck Models (CBNMs) [Koh2020], not against Concept Whitening [Chen2020].
>
> **Adding T to the notation in the experiments**.  Thank you for pointing this out.  We realize that we could have been more consistent in our usage of T.  We audited the paper and (1) added T in all equations related to leakage, as in this case the training and test distribution differ and this can be more clearly conveyed using T, see especially Sections 5.2, E and F; (2) left equations unrelated to leakage unchanged.
>
> **OSR setup is unclear**. We agree with you, our description could have been clearer. In short, prototypes are encoded in the data-driven prior p(z|y), which is modeled as a mixture of Gaussians. The complete procedure is as follows. The prior (and therefore the prototypes) are fitted jointly with the encoder, decoder, and classifier of GlanceNets, and specifically by optimizing the KL term in Eq 4.  Once learned, we use the means of the Gaussians as prototypes to carry out the OSR rejection step.  The procedure is adapted directly from [Sun2020].  We clarified the description in Section 4 and added further information on learning the prototypes in Section C.
>
> **Supervision on generative factors seems unrealistic**. As we mentioned in the paper, disentanglement (and therefore alignment) is not achievable in a fully unsupervised fashion without making non-trivial assumptions about the data generation process itself.  This is a well-known result first shown by [Locatello2019].  This, and the fact that CBNMs rely on concept-level supervision, motivates us to do the same.  We hope that this helps you see the motivation behind this choice.
>
> There may be situations in which supervision is not needed, but we tackle the general problem and leave a study of special cases to future work.  Moreover, concept annotations on 1%-25% of the examples are enough to ensure a reasonable amount of alignment.
>
> We agree that concept supervision is not always available.  We have listed in the paper that other forms of weak supervision can be used in place of dense concept annotations (see lines 221 to 224), such as partial concept annotations, grouping information, interaction with a human expert, etc.  All these strategies can be readily applied to VAE-based models like GlanceNets (but not to CBNMs).
>
> **There is no mixing of factors**. The reasons why we assume the generative factors G to be disentangled are (1) This mimics the typical XAI setting in which individual elements of an explanation are in some sense independent from one another to facilitate understanding; a similar rationale motivates the orthogonality criterion used in Concept Whitening; (2) There is a rich literature on disentanglement and many results that we could build on, such as the work of [Locatello2019]; (3) Work on more complex settings (e.g., hierarchical disentangled representations) is in its infancy.  As we mentioned in line 225, we plan to pursue this interesting direction in future work.
>
> **Multiple copies of monotonic transformation negatively affect alignment**. Good point. This is addressed by the DCI metric:  DCI wouldn’t be low even if different latent factors were monotonic transformations (different from each other) of the same $G_j$. We provide an example for that in Section B. (On the other hand, their Completeness *would* be low; please see [Eastwood2018] for a description of completeness vs. disentanglement in the DCI framework.)

---

> > ### Author Response · Authors · 2022-08-02
> > **Continuation of the reply to the Reviewer jG58**
> >
> >
> > **Human evaluation of learned concepts**. Human evaluations are useful in XAI to verify whether users understand explanations produced by a particular algorithm.  In our context, however, the added value of human evaluation is less clear.  What humans can do is determine whether a concept is semantically meaningful - typically by checking out if traversals in latent space make sense.  However, this form of evaluation is limited to a handful of images sampled from the decoder.  Our alignment metric instead systematically measures how good the learned representation is with the known generative factors on a very large test set.  In our context, it is not clear what his extra-human evaluation would contribute.  Regardless, we now do report concept traversals for GlanceNets in Section F to get anecdotical evidence of the kind of representations it learns. We reported the traversals for the TCVAE variant of GlanceNet with 10 supervised factors with 30 additional unsupervised ones, trained with full supervision on the concepts.
> >
> > **Applicability of GlanceNets and tasks with classes not determined by concepts**. In general, GlanceNets are well suited to handle applications that are currently dealt with using other concept-based models, like CBMs, SENNs, and ProtoPNets.  It is true that not all classification tasks admit a decomposition in terms of concepts, but we work under the assumption that this is the case - at least approximately.  This is useful to evaluate the performance of CBMs in situations where their inductive bias is most appropriate - such as when dealing with our chosen data sets, CUB200 [Chen2019], OAI [Koh2020], etc. - fairly against other CBMs, as we do here. Concerning all other data sets, we expect to achieve similar accuracy as CBMs even in situations where concepts are not enough.  Dealing properly with this case - in which the label strongly depends also on uninterpretable information - is left to future work.
> >
> > **Is OSR used in Section 5.1**? The aim of this Section is to compare the accuracy score and DCI of GlanceNets (which has a reject option) with CBNMs (which does not) evenly, so we disabled the OSR mechanism ni GlanceNets.
> >
> > **Size of** $\mathbf{Z}_{\bar{J}}$. In order to facilitate comparison with CBNMs, we chose  $\bar J = \emptyset$ in all experiments. We just included an additional experiment in Section F where we compare a CBNM with 10 latent dimensions (as in the original experiments) with (1) A GlanceNet (built on a TCVAE) with 10 supervised + 30 unsupervised latent dimensions, and (2) A GlanceNet (built on $\beta$-VAE) with 10 supervised + 10 unsupervised concepts. The results support our message that GlanceNets achieve better alignment and lower concept leakage than CBNMs while attaining comparable accuracy, and validate the ability of GlanceNets to deal with extra unsupervised latent variables.
> >
> > **Accuracy comparison with non-concept-based models**. Depending on the application, general deep networks tend to perform better than concept-based models in general, as they don’t need to acquire and make predictions with an interpretable concept vocabulary and don’t have a bottleneck.  For this reason, CBMs are generally compared against other interpretable approaches.  For instance, in [Koh2020] CBNMs are compared against (and shown to outperform) Self-explainable Neural Networks [Alvarez-Melis2018]. For the same reason, we compare against CBMs and show that we achieve similar prediction accuracy in practice.

---

> > > ### Author Response · Authors · 2022-08-06
> > > **Comment to Reviewer jG58**
> > >
> > > Dear Reviewer,
> > >
> > > We would like to follow up to see if our response addresses your concerns or if you have further questions. We would really appreciate the opportunity to discuss this further and know whether our response has already addressed your concerns. Thank you again!

---

> > > > ### Comment · Reviewer_jG58 · 2022-08-06
> > > > **reviewer reply to author rebuttal**
> > > >
> > > > Thank you for the time and effort you spent addressing my concerns and clarifying many points.  Please find my replies below where I still have concerns or am unclear.
> > > >
> > > > "Concept Whitening. Please note that we compare against Concept-bottleneck Models (CBNMs) [Koh2020], not against Concept Whitening [Chen2020].".
> > > >
> > > > - I'm a bit confused as in the paper on lines 236 and 237, you write " several tasks showing that GlanceNets outperform CBNMs [5]" and 5 refers to Chen2020 while 20 refers to Koh2020.  Am I still confusing this?
> > > >
> > > > "There is no mixing of factors. The reasons why we assume the generative factors G to be disentangled are (1) This mimics the typical XAI setting in which individual elements of an explanation are in some sense independent from one another to facilitate understanding; a similar rationale motivates the orthogonality criterion used in Concept Whitening; "
> > > >
> > > > - I'm not sure I entirely agree with this point since it seems to conflate the generative factors with explanations which in your case are the concepts and not the factors themselves, so its not exactly the same thing as wanting concepts to be orthogonal or something to the extent.
> > > >
> > > > "Applicability of GlanceNets and tasks with classes not determined by concepts. In general, GlanceNets are well suited to handle applications that are currently dealt with using other concept-based models, like CBMs, SENNs, and ProtoPNets. It is true that not all classification tasks admit a decomposition in terms of concepts, but we work under the assumption that this is the case - at least approximately. This is useful to evaluate the performance of CBMs in situations where their inductive bias is most appropriate - such as when dealing with our chosen data sets, CUB200 [Chen2019], OAI [Koh2020], etc. - fairly against other CBMs, as we do here. Concerning all other data sets, we expect to achieve similar accuracy as CBMs even in situations where concepts are not enough. Dealing properly with this case - in which the label strongly depends also on uninterpretable information - is left to future work."
> > > >
> > > > - In my original comment, I meant it would be interesting to have (i) N generative factors and 2*N concepts for instance , and (ii) it was a comment on the construction of the labels in Celebs dataset use.   For (i) you can still decompose the the classification task in terms of concepts, its just not one to one .
> > > >
> > > > "Size of Zj. In order to facilitate comparison with CBNMs, we chose J_bar=0 in all experiments."
> > > > - I think its good what you did in Appendix F, but having Z_J_bar be empty for all experiments seems to imply you don't need that part of your architecture, even though its what you use for the OCR filtering step no?  I think this part needs to be made very explicit in your experiment setup and justified as you do.
> > > >
> > > > "Accuracy comparison with non-concept-based models. Depending on the application, general deep networks tend to perform better than concept-based models in general, as they don’t need to acquire and make predictions with an interpretable concept vocabulary and don’t have a bottleneck. For this reason, CBMs are generally compared against other interpretable approaches. For instance, in [Koh2020] CBNMs are compared against (and shown to outperform) Self-explainable Neural Networks [Alvarez-Melis2018]. For the same reason, we compare against CBMs and show that we achieve similar prediction accuracy in practice."
> > > >
> > > > -   CBNMs are compared at the concept accuracy level against Alvarez-Melis2018, but at the classification task level, I don’t think its true that general deep networks necessarily perform better than concept-based models in general ( such as for Prototype work (Li 2018) ).  I do think it would be interesting to show how simplified the assumptions of these factor-concept linking affects task performance ( in particular with regard to the size of Z_J_Bar )
> > > >
> > > > Overall I think the clarifications added to the paper and the additional experiment in section F have helped some with clarifying the paper ( and I have revised my score), but my main concerns on human eval and utility of the simplified assumption still remain and could be used to strengthen future work in this interesting area

---

### Official Review · Reviewer_31JG · 2022-07-11

**Rating:** 6
**Confidence:** 3
**Soundness:** 3 good
**Presentation:** 3 good
**Contribution:** 3 good

**Summary:**

The paper defines a model that the authors claim to be a concept based model that solves two key issues in the domain of concept based models and interpretable models more widely. The first is that the concepts are demonstrably interpretable. Here the authors present a (rather neat) definition of interpretability in terms of alignment, where:
> learned concepts are interpretable if they can be mapped to a (partially) interpretable data generation process using a transformation that preserves semantics.

This is a powerful notion and the mathematical arguments around sufficient conditions for alignment. Their model,training process and evaluation metrics are then informed by these. There are some additional arguments relating to disentanglement, but these are less clear to me.

The paper presents a model that combined a disentangled (beta-)VAE model with an additional concept decoder for a subset of supervised examples on the (otherwise) hidden factors. The auxiliary loss is described as an average cross entropy loss where (for reasons that aren't entirely clear to me):
> the annotations g k are rescaled to lie in [0, 1]

The paper also presents an argument about concept leakage. The definition of this is a little hazy but at the  first presentation it is described as:
> whereby the learned concepts end up encoding spurious information about unrelated aspects of the
data, making it hard to assign them clear semantics.

Their model addresses concept leakage by modelling the distribution of learned interpretable factors and rejecting predictions at test time that are outliers for this distribution. In the evaluations later in the paper, these rejections are implemented by making random predictions. However this is more to meet their evaluation requirements than a suggestion of what to do with outlier predictions.

The paper then presents a series of evaluations comparing the proposed model (GlanceNets) and a pre-existing concept model (CBNM). These evaluations involve 3 datasets, two pre-existing synthetic benchmarks for disentangled and related models: dSprits and MPI3D, and an image dataset of celebrities (CelebA), labelled with "interpretable features" by crowdsourced workers. The two datasets MPI3D and Celeb were then given labels based on clustering the "interpretable factors".

Performance is measured by three measures, accuracy, alignment (as defined by the authors) and explictness (predictive performance on hidden/interpretable features). The proposed model outperforms (or is competitive with) CBMN on all three metrics on all three datasets. In particular, GlanceNets outperforms significantly on the alignment metric for the synthetic datasets.


**Questions:**

The leak proof experiments are a little unclear. As I read it, the representations are trained on 4s and 5s from the MNIST dataset. Then, I am guessing, a linear classifier is trained on fixed embeddings for all digits 0-9, where the target label is binary (odds/evens). If there is consistent information in the representation to allow the prediction to be made then it will be above 0.5. Is that right?

I have also tried to outline my understanding in the strengths and weaknesses section. In terms of these interpretations have I misunderstood anything?

**Limitations:**

There could be a little more on the limitations of the approach and the limitations of the evaluations.

**Strengths And Weaknesses:**

The paper presents a novel and, in my opinion, convincing argument that interpretability can be defined in terms of *alignment* with interpretable features. The subsequent mathematical arguments relating to alignment are, mostly, very clear and well argued. Equally the argument that the existing DCI metric (indirectly) measures alignment.

The arguments about concept leakage are less clear, and would really benefit from a much clearer, upfront, definition of concept leakage alongside an example. There are also other issues of clarity, including on disentanglement and some of the descriptions of their model/evaluations (see below).

Along with the issues with clarity I have some concerns about the strength of the evaluations. The results do show that the GlanceNet model improves alignment with the features/factors that are said to be interpretable, but the datasets chosen are either entirely synthetic (dSprites and MPI3D), and/or have labels that are constructed from some synthetic construction (MPI3D and CelebA). As such there is a danger that these results are confirming the authors preconceptions to some degree. At this level, I would expect to see at least one dataset where there hasn't been intervention in the data generation process. I realise the challenge that this presents researchers in this (and related fields) but it is a principle I feel is important for the advancement of the field. For instance, some kind of human study that evaluates the interpretability of learn features would be the gold standard in my opinion. For an excellent example see (Chang et al., 2009) - details below.

Another minor issue I have is with the fairness of the experiment showing concept leakage. Really, the authors are making a claim that out of sample test predictions should be rejected if the interpretable/latent factors fall outside of the centre of mass of points seen at training time. This is an excellent observation and not one that I intend to dispute. Nontheless, the evaluation then sets up their method with a (quasi-)rejection option based on this principle and compare it to methods that do not have such a reject mechanism. I don't know what a fair evaluation would look like here, but this doesn't meet those requirements in my opinion. It also feels a little contrived that the reject option is to make a random prediction. Were these random predictions included in the earlier experiments too?

So, in summary, I like this paper, and I am convinced by the main causal argument, but I think that there are some outstanding issues that mean I cannot recommend it for acceptance as it stands.


I have a few additional comments about details in the paper that came up as I was reading. These are as follows:


The authors claim about concept leakage that:
> This phenomenon, illustrated in Fig. 1 is known as concept leakage.

However, the image seems more to demonstrate how their model avoids concept leakage rather than the phenomenon itself. The authors then say (about the literature) that:
> a clear definition of leakage is missing

Then about their paper that:
> A key contribution of our paper is showing that leakage can
be understood from the perspective of domain shift and dealt with using open-set recognition

But this feels like a bit of a non-sequitur. A definition should be present and unambiguously described before it can be understood from different perspectives. In the absence of a definition then a better intuitive description would help, with some clear examples.

There is a similar issue with the definition of disentanglement. I realise that this is one of the key issues in the disentanglement literature, that there are disputed terms, but, perhaps because of this, it is important that a working definition is given at the very least. The paper later associates semantically preserving maps with something *stronger than disentanglement* but do not give a clear definition of tihs either.

On line 133 you describe what appears to be a row vector left multiplied by a matrix. Elsewhere it could be clearer where vectors are defined that they are treated as column vectors.

This sentence (line 308-310) doesn't seem to parse very clearly:
> the goal is to discriminate between even and odd MNIST images using a latent representation
Z = (Z 4 , Z 5 ) obtained by trained (with complete supervision on the generative factors) only on
examples of 4’s and 5’s.

---

> ### Author Response · Authors · 2022-08-02
> **Reply to Reviewer 31JG**
>
> Thank you for taking the time to review our work.  We appreciated that you noticed how our paper tackles two important open problems and that it supplies a powerful notion of interpretability.  We will update the text to take care of all the minor issues you reported. Please find below detailed replies to the major points in your review.
>
> **Definition of concept leakage**. Thank you for pointing this out. We have extended our description of concept leakage in Section 3 and remarked that it can be understood as the presence of both (i) some factors of variations that do not change in the training phase ($\mathbf G_F$ in the main text) and (ii) correlations among the two group of factors ($\mathbf G_V$ and $\mathbf G_F$ in the main text).  We hope that now it is more clear.
>
> **Definition of disentanglement**. There is no unique, agreed-upon definition of disentanglement in the literature, but formal and well-known definitions include those of [Eastwood2018, Suter2019, Shu2020]. Our definition of disentanglement, now given in line 128 (in item (i)), is essentially the same as [Suter2019], and alignment specializes this notion with an additional element-wise monotonicity constraint. Notice that we do not measure disentanglement in our experiments, but rather alignment, which we define within the DCI measurement framework of [Eastwood2018]. Other metrics for disentanglement, such as IRS for instance [Suter2019], neglect monotonicity and cannot be trivially adapted for computing alignment.
>
> **Datasets are synthetic**. We considered using standard data sets used to evaluate other concept-based models, specifically, the CUB bird classification data set, which comes with concept-level annotations.  However, CUB is extremely challenging for reconstruction, as it encompasses 200 classes of real-world images with ~40 examples per class.  We opted for complex but larger data sets from the disentanglement literature, that are designed for generative models and therefore have no label supervision. These three data sets are very reasonable. dSprites and MPI3D were chosen because they are frequently used for disentanglement and exhaustive, facilitating evaluation of DCI on their large test sets. (They are a common testbed for disentanglement research.) On the other hand, while CelebA contains a bigger number of training instances w.r.t. CUB, it does not display all variations of the annotated factor, nor they are exhaustive to describe the whole image. CelebA is not exhaustive and as such quite challenging.
>
> **Human evaluation of learned concepts**. Human evaluations are useful in XAI to verify whether users understand explanations produced by a particular algorithm.  In our context, however, the added value of human evaluation is less clear.  What humans can do is determine whether a concept is semantically meaningful - typically by checking out if traversals in latent space make sense.  However, this form of evaluation is limited to a handful of images sampled from the decoder.  Our alignment metric instead systematically measures how good the learned representation is with the known generative factors on a very large test set.  In our context, it is not clear what his extra-human evaluation would contribute.  Regardless, we now do report concept traversals for GlanceNets in Section F to get anecdotical evidence of the kind of representations it learns. We reported the traversals for the TCVAE variant of GlanceNet with 10 supervised factors with 30 additional unsupervised ones, trained with full supervision on the concepts.

---

> > ### Author Response · Authors · 2022-08-02
> > **Continuation of the reply to Reviewer 31JG**
> >
> > **Comparing against competitors with no rejection mechanism is unfair**. We politely disagree.  Dealing with concept leakage using an OSR rejection mechanism is one of the two key contributions of our paper (the other being a definition of interpretability as alignment) and the goal of our leakage experiments is to validate whether this is useful or not.  To do so fairly, we compare GlanceNets against CG-VAEs, which are GlanceNets with the rejection mechanism removed.  The results of this *ablation* experiment show that rejection helps tremendously.  We also report results for CBNMs, although they are not the most direct competitor.  This is useful to highlight that CBNMs, as shown by [Mahinpei2021] and [Margeloiu2021], are susceptible to leakage.  We will clarify this in the text.
> >
> > **Why does rejection yield a random prediction**. The intuition is that, if an instance $x$ is rejected, we cannot really trust its latent representation $z$.  In absence of this crucial information (and without building a full-fledged alternative classifier that maps x to y), the best way to predict a reasonable label is to draw a random prediction from the prior p(y).  The latter is generally unknown for the test data, but hopefully, a reasonable estimate can be guessed based on domain knowledge - or just set to the uniform distribution whenever no domain knowledge is available.  In our case, the two options coincide, as all labels are equally likely in the test set. Of course, other options are possible, and one can implement the reject option as they see fit.
> >
> > **GlanceNets avoid leakage rather than the phenomenon itself**. OSR deals with leakage at *inference time*, not at training time - so it cannot (and needs not to) affect the learned concepts.  Notice that the difference between GlanceNets and VAEs lies in the prior (which in our case is a mixture of Gaussians and implicitly clusters the learned concepts using prototypes) and in the OSR rejection step (which relies on the prior), rather than in the learned representation.  We stressed that OSR is an inference-time strategy at the end of Section 3.
> >
> > **Vectors and transposition**. Thank you for noticing this! We updated the manuscript to make the notation more uniform.
> >
> > **Design of the MNIST leakage experiment**. The experiment is set up as suggested in [Mahinpei21] and discussed in Section 2 (lines 79-91):  first, we learn a GlanceNet or CBNM encoder (and decoder) using concept supervision about what examples are 4’s or 5’s but no label supervision.  So there is no downstream task in this step.  In a second step, we train a linear classifier on top of the concepts learned in the previous step, using even vs odd labels.  We reported this information in Section D.

---

> > > ### Author Response · Authors · 2022-08-06
> > > **Comment to Reviewer 31JG**
> > >
> > > Dear Reviewer,
> > >
> > > We would like to follow up to see if our response addresses your concerns or if you have further questions. We would really appreciate the opportunity to discuss this further and know whether our response has already addressed your concerns. Thank you again!

---

> > > ### Author Response · Authors · 2022-08-08
> > > **Message to Reviewer 31JG**
> > >
> > > Dear Reviewer 31JG,
> > > considering that the author-reviewer discussion period is coming to a close, we politely ask if the rebuttal we provided has managed to answer your questions.  We'd be happy to answer any additional queries you might have.
> > > Best regards,
> > >
> > > The authors

---

### Official Review · Reviewer_SNnk · 2022-07-18

**Rating:** 6
**Confidence:** 4
**Soundness:** 3 good
**Presentation:** 3 good
**Contribution:** 3 good

**Summary:**

This paper present GlanceNets as a new approach for learning interpretable DNNs whose presentations maps to a set of human aligned concepts. The primary improvement that comes with using GlanceNets is that they do not suffer from concept leakage, which is a challenge for predecessors that tried to address this problem. The GlanceNet is a VAE trained to represent 1) interpretable concepts, 2) Non-interpretable concepts, and 3) an out-of-distribution head to reject samples at test-time. Empirical assessment of the GlanceNets shows that they are not susceptible to leakage and competitive on accuracy.

**Questions:**

The discussion around the questions I have is provided in the weakness bullet list in the previous section. Here I give a shortened version of that discussion:

- Can you give an example, in terms of the 4/5 classification problem, how the concept leakage problem is a domain shift one? Specifically, what is $\mathbf{G_V}$ in this problem? Is it the 4 and 5 concept? If yes, what is $\mathbf{G_F}$? Is it any other concept that can be used to classifier examples that are not 4/5?

**Limitations:**

No explicit discussion of the limitations of this work in the draft. The authors can add this towards the end of the paper.

**Strengths And Weaknesses:**

## Strengths

- **Tackles an important problem**: This paper addresses the important concept leakage problem, and demonstrates convincingly that the the proposed architecture addresses the problem. In addition that paper shows that the model is competitive on accuracy (figure 3), while achieving better alignment. Section 5.2 critically demonstrates that Glancenets do not suffer from concept leakage.

- **Intuitive Rethinking of Interpretability as alignment**: One of the key contributions of this work is its focus on the alignment problem. Specifically, the paper seeks to make sure that the concepts relied on by a classifier are 'aligned' with the concept representations provided by a domain expert. Section 3 provides an interesting discussion on this front, which could likely be adapted by future work in this area.

## Weaknesses
- **The Interpretability as Concept leakage paragraph**: This is the key paragraph that justifies the proposed architecture in my opinion, but I don't follow the logic here. Specifically, why is the issue a domain shift problem? Couldn't it still be possible to train a CBNM where there is not domain shift but there is still concept leakage? Perhaps the authors can use the 4/5 classification example as an example here.

- **It seems like the solution here punts the problem**: As far as I am aware, following the 4/5 example, representations learned from the 4/5 classification are effective for classification other instances. However, to tackle this issue this paper suggests adding a component that rejects any input that is not a 4/5, but it is unclear as whether there is still concept leakage in the representations. Specifically, my question is whether there is still leakage if you don't reject out of domain examples? I say this because it seems to me that ood identification and the leakage/alignment problem here are somewhat orthogonal. Given the definition of alignment that this paper provides, it would've been great if the paper showed that optimizing for that objective leads to a more aligned representation for glancenets.

- **Notation**: This paper has a ton of notation that makes it somewhat difficult to digest. Specifically, $\mathbf{G_I}$, $\mathbf{g_I}$, and $g_I$ somewhat interchangeably. It'll help to unify this notation.

---

> ### Author Response · Authors · 2022-08-02
> **Reply to Reviewer SNnk**
>
> Thank you for your review, we are happy that you noticed the novelty and potential impact of the work. We report below our replies to the points you raised.
>
> **Unclear why concept leakage is domain shift**.  We agree that our description should be improved.  Consider the even vs odd MNIST task.  Now, only images of 4’s and 5’s are observed when training the encoder (step 1), all other concepts having probability zero, and all other digits appear when training the classification layer (step 2; with concepts 4 and 5 now having probability zero).  The distribution over concepts clearly changes when moving from task (1) to task (2).  Now, what happens in concept leakage is that the concepts learned in (1) correlate with those observed in (2), which are necessary to solve the even vs odd classification task, yielding non-random prediction accuracy.  This complicates interpretability.  But if we knew that an incoming instance includes a concept in (2) that has never been observed during training, we could reject it, rather than leaking information about the concepts in (1).  This is exactly what we accomplish with the open-set recognition step.  Hopefully, this clarifies our proposal.
>
> **Is concept leakage possible without domain shift?** No, by construction.  In our description of leakage, we specify that $G_F$ is fixed in the training set (when $T=1$) but not in the test set ($T = 0$).  This is a form of domain shift.
>
> **Is there still leakage without rejection?** Yes.  As you can see from Figures 4 and 5, the CG-VAE (= GlanceNets without the OSR component) is affected by high levels of concept leakage.
>
>
> **Show that OSR improves the alignment of representations**. OSR deals with leakage at *inference time*, not at training time - so it cannot affect the learned concepts.  Notice that the difference between GlanceNets and VAEs lies in the prior (which in our case is a mixture of Gaussians and implicitly clusters the learned concepts using prototypes) and in the OSR rejection step (which relies on the prior), rather than in the learned representation.  We stressed that OSR is an inference-time strategy at the end of Section 3.
>
> **Notation**.  Thank you for pointing this out. We will go through the paper and streamline all notation.

---

> > ### Author Response · Authors · 2022-08-06
> > **Comment to Reviewer SNnk**
> >
> > Dear Reviewer,
> >
> > We would like to follow up to see if our response addresses your concerns or if you have further questions. We would really appreciate the opportunity to discuss this further and know whether our response has already addressed your concerns. Thank you again!

---

> > > ### Comment · Reviewer_SNnk · 2022-08-08
> > > **Thanks for the response**
> > >
> > > Thanks for the ping and the response. The response clarifies some of the confusion that I had. The only remaining piece for me is why the solution to concept leakage, at test time, should be to reject samples. It seems like the work is saying the key way to avoid concept leakage is to reject samples based on how similar the latent code of the test predictions are to those observed during training. This strategy is effective here, but leaves me puzzled as whether the solution is really just side-stepping the issue. With all this said, happy with the author response.

---

### Author Response · Authors · 2022-08-04
**Updated pdf and code**

Hi,
this is a heads up that we just updated the pdf and the code of our submission.
Best regards,

The authors

---

### Meta-Review · Area_Chair_2ruo · 2022-08-25

**Recommendation:** Accept
**Confidence:** Certain

**Metareview:**

I thank the authors for their submission and active participation in the discussions. The paper presents a method for interpretable deep learning. All reviewers unanimously agree that this is a solid paper worthy of acceptance. In particular, reviewers noted that the paper tackles an important problem [SNnk], and presents a novel [jG58,Xfpf], interesting [SNnk,mGx1] and intuitive [31JG] framing of interpretability as alignment problem. Furthermore, the writing is clear [jG58,Xfpf] and the empirical validation extensive [Xfpf]. Thus, I am recommending acceptance of the paper and encourage the authors to further improve their paper based on the reviewer feedback.

**Award:**

No

---

### Decision · Program_Chairs · 2022-09-14

Accept